# Promoter-enhancer interactions identified from Hi-C data using probabilistic models and hierarchical topological domains

Gil Ron[1], Yuval Globerson[1], Dror Moran[1] & Tommy Kaplan [1]

Proximity-ligation methods such as Hi-C allow us to map physical DNA–DNA interactions along the genome, and reveal its organization into topologically associating domains (TADs). As the Hi-C data accumulate, computational methods were developed for identifying domain borders in multiple cell types and organisms. Here, we present PSYCHIC, a computational approach for analyzing Hi-C data and identifying promoter–enhancer interactions. We use a unified probabilistic model to segment the genome into domains, which we then merge hierarchically and fit using a local background model, allowing us to identify over-represented DNA–DNA interactions across the genome. By analyzing the published Hi-C data sets in human and mouse, we identify hundreds of thousands of putative enhancers and their target genes, and compile an extensive genome-wide catalog of gene regulation in human and mouse. As we show, our predictions are highly enriched for ChIP-seq and DNA accessibility data, evolutionary conservation, eQTLs and other DNA–DNA interaction data.

[1] School of Computer Science and Engineering, The Hebrew University of Jerusalem, Jerusalem, 9190401, Israel. Correspondence and requests for materials should be addressed to T.K. (email: tommy@cs.huji.ac.il)

One of the key mechanisms of gene regulation in eukaryotes involves promoter–enhancer interactions, where distal regulatory regions along the DNA (enhancers) come in close physical proximity to their target promoters to further activate transcription. The human genome is estimated to contain hundreds of thousands of enhancers, often with multiple enhancers regulating a single gene. These act in a tissue-specific manner and could be found up to 1 Mb away from their target genes[1–6]. The importance of enhancers for gene regulation is further emphasized by a growing body of works that link genetic variation in enhancer sequences to human diseases[7–11]. Nonetheless, we still lack a deep understanding of the following: (a) how enhancers work molecularly, (b) how their tissue specificity is encoded in their sequence, and above all, (c) how they recognize and physically interact with their target genes.

In recent years, high-throughput molecular methods have been developed to study the three-dimensional organization of the genome, and its relation to various functions. For example, proximity-ligation methods such as 4C, ChIA-PET and Hi-C quantify the frequency of DNA–DNA interactions in living cells and map the 3D organization of the genome in high resolution[12–23]. To date, Hi-C experiments were performed in a variety of organisms and cellular conditions, including many cell types and tissues.

While the genomic resolution of these data is often low, varying from few Kbs to 40Kb blocks, they were mainly used to identify and delineate topologically associating domains (TADs). These are continuous regions (hundreds of Kbs to few Mbs) that were shown to be folded upon themselves into local compartments and facilitate high number of DNA–DNA interactions[19,24–26].

In recent years, topological domains were studied extensively, and were shown to be (a) related to replication domains[27,28], (b) largely conserved across evolution, and (c) play a crucial role in chromosome function[25,29–33].

TADs also play a key role in gene regulation, as they define the regulatory scope of enhancers. The domains' boundaries were shown to act as regulatory "insulators" that prevent targeting genes outside of the enhancer domain[34,35]. Disruptions of the chromosomal structure, either in human genetic disorders or by artificially deleting boundary elements (e.g., using CRISPR-Cas9), were shown to be associated with enhancer mis-regulation and aberrant gene expression[9–11,36–38]. While we still lack a deep understanding of the exact mechanisms by which topological domains are defined and maintained, TAD borders were shown to be enriched for highly transcribed genes[25], as well as CTCF and cohesin binding sites[22,31,39–45].

As more and more 3D data accumulate, in a multitude of tissues and cellular conditions, algorithms were developed to analyze Hi-C data and partition the genome into a set of topological domains[17,20,25,46–50]. Most notable are the Directionality Index method[25] that scans the genome by analyzing the set of DNA–DNA interactions for every locus, and identifies transitions from loci with mostly backward interactions to adjacent loci with mostly forward interactions; and the Insulation Square method[23] that identifies TAD boundaries as genomic loci with very few overhead interactions. Additional methods aim to construct a more hierarchical structure of topological domains, a visible feature of Hi-C maps, either by merging cross-connected sub-domains into larger domains[20] or by iteratively altering the algorithm parameters to obtain an ensemble of multiple chromosomal segmentations that could be interpreted as hierarchical domains[50]. While these methods are generally fast and robust, they are inherently biased towards short-range interactions that form the vast majority of DNA–DNA interactions, thus shading

the less abundant long-range interactions (250 Kb and above), that are more informative for calling hierarchical TADs.

Here, we present PSYCHIC (Fig. 1)—a three-step modular algorithm to identify promoter–enhancer interactions. Briefly, we use a unified probabilistic model and a Dynamic Programming algorithm to find an optimal segmentation of each chromosome into topological domains; we next iteratively merge neighboring domains into hierarchical structures; and finally we fit each domain using a local background model. This allows us to identify over-represented DNA–DNA pairs, including enhancers and their target genes. We have analyzed the Hi-C data from 15 conditions and cell types in mouse and human[19,20,25], and identified hundreds of thousands of over-represented interactions. This comprehensive genome-wide tissue-specific database of putative interactions between enhancers and their target genes would be of great interest to the scientific community.

## Results

**A unified probabilistic mixture model for Hi-C data.** Hi-C interaction maps often show a clear distinction between two different patterns—Rectangular regions along the diagonal of the Hi-C map that correspond to topological domains, and present high intensity of (intra-domain) DNA–DNA interactions. These are often surrounded by regions with fewer (inter-domain) DNA–DNA interactions. Due to symmetry, Hi-C maps are often rotated in 45 degrees, with topological domains shown as isosceles right triangles along the (now horizontal) diagonal of the Hi-C map (Fig. 1a).

We begin by developing a simple two-component probabilistic model, corresponding to the probability of intra- and inter-TAD interactions. In brief, our algorithm analyzes the Hi-C interaction matrix and infers for every cell (DNA–DNA pair) the log-probability ratio (LPR) of these loci occurring within the same topological domain or not. In the following stages, we will combine these ratios into a unified score, and use Dynamic Programming to optimally segment each chromosome into domains.

Formally, let $P_d(N)$ denote the probability of observing $N$ Hi-C interactions between two DNA loci $d$ bases apart. This equals to the weighted sum of the intra-domain and inter-domain sub-models:

$$P_d(N) = P_d(\text{intra}) \cdot P_d(N|\text{intra}) + P_d(\text{inter}) \cdot P_d(N|\text{inter}) \quad (1)$$

where $P_d(N|\text{intra})$ and $P_d(N|\text{inter})$ correspond to the likelihood of observing $N$ interactions $d$ bp apart in the intra-TAD and inter-TAD sub-models, respectively. $P_d(\text{intra})$ and $P_d(\text{inter})$ correspond to the a priori probability of observing two loci $d$ bp apart to be within or outside of the same TAD. For robustness, we model $N$ using a log-Normal distribution (Supplementary Fig. 1a, b; Methods section). Additional probabilistic families (log-Poisson and Negative Binomial) were considered and found to be less accurate (Supplementary Fig. 1c, d). This parameterization greatly reduces the number of free parameters, resulting in a compact model $\theta_d$ with only six parameters for every distance $d$, including $\mu_d^{\text{intra}}$, $\sigma_d^{\text{intra}}$, $\mu_d^{\text{inter}}$, and $\sigma_d^{\text{inter}}$ (mean and standard deviation parameters for intra-TAD and inter-TAD models); and two prior parameters $P_d(\text{intra})$ and $P_d(\text{inter})$, while offering an accurate approximation of the Hi-C data (Supplementary Fig. 1a, b). For every distance $d$, we directly estimate the model parameters from annotated Hi-C data: To estimate $\theta_d$, we rely on an initial (possible noisy) segmentation of the Hi-C map into domains. These could be obtained using various methods, including the directionality index (DI) HMM-based method of Dixon et al[25], Insulation Square[23], or approximated iteratively using the Expectation-Maximization (EM) algorithm[51]. Given

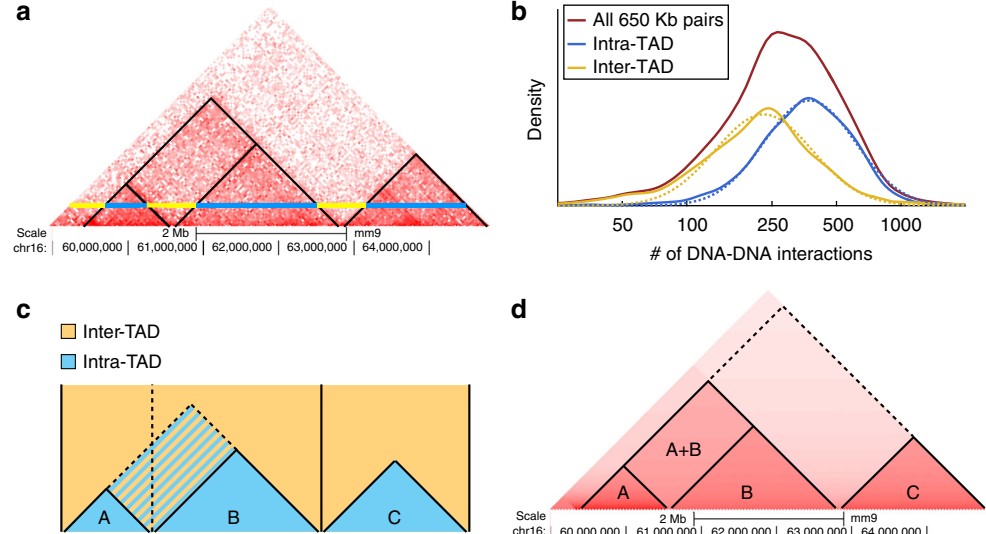

**Fig. 1** Overview of the PSYCHIC algorithm. **a** Example of Hi-C interaction map (rotated in 45°), from mouse cortex (chr16, 59–65 Mb)[25]. Blue and yellow horizontal lines correspond to DNA–DNA pairs, 650 Kb apart, within and across domains. **b** Histograms show the empirical abundance of these DNA–DNA interactions, either within domains (blue) or across domains (yellow), and demonstrate the enrichment of intra-TAD interactions. Dotted lines show a log-Normal distribution fitted to these empirical data. **c** PSYCHIC first uses a two-component probabilistic mixture model to estimate the number of intra-TAD (blue) and inter-TAD (yellow) DNA–DNA interactions. For example, shown is segmentation into three domains A–C (delineated by vertical lines). An alternative segmentation, where A and B domains are unified now consider the striped rectangle as intra-TAD. PSYCHIC uses a log-posterior ratio score with a Dynamic Programming algorithm to identify the optimal (Viterbi) segmentation of the chromosome into domains. **d**. PSYCHIC then iteratively merges similar neighboring domains (here, A + B) into hierarchical structures. For example, dotted lines marks a possible 2^nd-order merge between the merged (A + B) domain and domain C. PSYCHIC then fits a bi-linear power-law model for each TAD or merge to reconstruct a domain-specific background model (shown by different shades of red). This allows for the identification of over-represented DNA–DNA pairs, including putative promoter–enhancer interactions

such annotations, we consider all intra- and inter-TAD pairs and use a maximum likelihood estimation of the mean and the standard deviation parameters. As shown by comparing different chromosomes of mouse ES cells, these estimations are very robust (Supplementary Fig. 1e). The same approach is used to estimate the prior probabilities, namely which percent of the DNA–DNA interactions of distance $d$ occur within, or across, topological domains.

**Identification of TAD boundaries using log-posterior ratios**. Using the above probabilistic model, we now wish to re-segment the genome into domains. For this, we propose a score that will integrate information from various distances of DNA–DNA interactions across the entire Hi-C matrix, without being skewed by the significantly higher number of interactions among nearby DNA–DNA pairs.

For this, we define a local score that calculates for every cell in the Hi-C matrix the log-posterior ratio (LPR) of the intra- and inter-TAD sub-models. Assuming $N$ interactions for two DNA loci $d$ bases apart, we could use Bayes' law to derive the posterior probability of being within $P_d(\text{intra} \mid N)$ or between TADs $P_d(\text{inter} \mid N)$ (Methods section). This allows us to compute the log-posterior ratio of the two sub-models:

$$\text{LPR}_d(N) = \log \frac{P_d(\text{intra} \mid N)}{P_d(\text{inter} \mid N)} \quad (2)$$

We are now ready to score a segmentation of the genome into domains.

First, let us define the probabilistic score for a single topological domain $t$, starting at position $s$ and ending at position $e$. For this, we sum the log-posterior ratios for all intra-TAD cells (pairs $<i,j>$ such that $s \leq i \leq j \leq e$), and subtract the log-posterior ratios for

all inter-TAD cells outside of TAD $t$. These are defined by the remaining (non intra-TAD) pairs $<k,l>$ whose centers lie within the TAD $t$, such that $s \leq (k + l)/2 \leq e$.

$$S(t) = \sum_{<i,j> \in t} \text{LPR}_{|j-i|}(N_{i,j}) - \sum_{<k,l> \notin t} \text{LPR}_{|l-k|}(N_{k,l}) \quad (3)$$

These are shown as blue (intra-) and yellow (inter-TAD) regions in Fig. 1c. For efficiency reasons, we only consider intra-TAD pairs ($<i,j>$) or inter-TAD ($<k,l>$) up to a maximal distance $h$ of 5 Mb. Probabilistically speaking, we allow every Hi-C cell to independently compare its likelihood given each of the two sub-models. We then define a global score for a segmentation $C$ of the genome into a set of TADs, by summing over their respective scores:

$$\text{Score}(C) = \sum_{t \in C} S(t) \quad (4)$$

As shown in Fig. 1c, the score of each TAD $t$ is based on pairs within $t$ (blue) or directly above $t$ (yellow), such that all Hi-C cells are counted exactly once. Moreover, since the score is strictly additive, breaking a single TAD into two TADs requires to only change the sign of LPR scores for cells between those TADs (Fig. 1c, striped region), as they are shifted from being considered intra-TAD (thus positive, left-hand side of Eq. 3) to inter-TAD (negative, right-hand side of Eq. 3).

Finally, we use a Dynamic Programming algorithm to find the optimal segmentation of each chromosome into topological domains, with respect to our two-component model. For this, we use a Dynamic Programming algorithm that computes the optimal score of each genomic interval $C_{i,j}$ by comparing its score as a single TAD from position $i$ to position $j$, $S(t_{i,j})$ as in Eq. 3, or by recursively breaking it at each possible position $k$, into two

distinct regions, one ranging from position $i$ to $k$, and another region from position $k + 1$ to position $j$:

$$\text{Score}\,(C_{i,j}) = \max_{i<k<j} \begin{cases} S(t_{i,j}) \\ \text{Score}\,(C_{i,k}) + \text{Score}\,(C_{k+1,j}) \end{cases} \quad (5)$$

Our algorithm then extends the computed range <$i,j$> until the entire chromosome is covered. This allows us to efficiently enumerate over all possible configurations {$C$} for each chromosome and identity the optimal segmentation $C$, with respect to the above probabilistic score.

**Hierarchical model of topological domains.** So far, we developed a probabilistic framework for modeling the Hi-C data within and across topological domains, and presented an efficient algorithm for identifying the optimal segmentation. For this, our model assumed that all intra-TAD DNA–DNA pairs, located $d$ bases apart, distribute according to one set of log-Normal parameters, and all inter-TAD pairs use another set.

We now wish to alleviate this assumption, and allow each TAD to fit a unique set of parameters fitting its intra-TAD Hi-C interaction counts. In addition, we wish to fit additional sets of parameters to selected inter-TAD regions (shown as tilted rectangles in the Hi-C map, Fig. 1c).

Specifically, we wish to iteratively agglomerate neighboring TADs into hierarchical structures of topological domains, where each TAD or merged regions is assumed to have a different tendency for Hi-C interactions (Fig. 1d). For this, we developed a "merge score" that allows us to examine adjacent domains. A naive scoring system for neighboring TADs would simply quantify their connectivity, by directly counting the number of inter-TAD interactions[20]. This score, however, might be biased by the size of the two domains, as well as the overall interaction intensity in each of the two domains.

Instead, our "merge score" preferentially chooses neighboring TADs whose inter-TAD region is more similar to each of the intra-TAD regions than to the overall inter-TAD Hi-C count distributions. Specifically, we calculate for each domain the average number of DNA–DNA interactions at any distance $d$ (Supplementary Fig. 1f), and compare these plots to the region between the two TADs, and to the remaining inter-TAD regions ("Sky" in Supplementary Fig. 1f). We then linearly regress these plots, and find the optimal $\alpha$ satisfying:

$$I_{\text{Merge}}(d) \approx \alpha \cdot I_{\text{TADs}}(d) + (1 - \alpha) \cdot I_{\text{Sky}}(d), \quad (6)$$

where $I_{\text{Merge}}$, $I_{\text{TADs}}$, and $I_{\text{Sky}}$ denote the average intensities for each $d$ at the inter-TAD ("Merge") area, the intra-TAD interactions within the two TADs, and the inter-TAD background model ("Sky"). We do so iteratively, greedily merging TAD pairs with the highest $\alpha$ value. Specifically, we merge two adjacent TADs whose inter-TAD region is the most similar (in terms of Hi-C interactions) and most dissimilar to inter-TAD regions. As before, this is done iteratively up to a maximal merge size of 5 Mb, to create a set (forest) of tree-like TAD merges, visually corresponding to triangles (TADs) and rectangles (inter-TAD merge regions). Supplementary Fig. 1g compares the number of TADs and hierarchical merges (1st, 2nd order, etc) for various Hi-C data sets.

**TAD-specific background model using Bi-linear power-law fit.** Once we segmented the Hi-C map into topological domains and TAD merges, we wish to specifically model the intensity of Hi-C data in each region, thus fitting the Hi-C data with a series of local background models. This will allow us to estimate the expected number of interactions in each Hi-C cell, thus

identifying over-represented Hi-C cells enriched compared to their specific TAD environment. Previous works used a power-law scaling model[15,52,53] to regress the expected number of DNA–DNA interactions as a function of their distance $d$:

$$I(d) \propto d^a \quad (7)$$

This is often plotted in log–log scale, where the (log) number of interactions scales linearly with the (log) distance:

$$\log(I) = a \cdot \log(\Delta) + b \quad (8)$$

with $a$ being the power-law coefficient (slope of log–log plot) and $b$ is the intersection parameter.

Nonetheless, while we found the power-law model to be generally accurate, it is clear that some domains show more Hi-C interactions than others (Fig. 1a), suggesting they would be best described by different power-law parameters (Supplementary Fig 1f, e.g. TADs A vs. B). We therefore wish to fit a different background model for each TAD and each merged region (Fig. 1d). This allows us to estimate the expected number of interactions at any distance within every topological domain/merge and quantify the statistical significance of over-represented interactions.

Next, we quantified the goodness-of-fit of each model to Hi-C data (Supplementary Fig. 2). First, we tested the overall fit with a single model for each chromosome, yielding an average RMSE of 1.45. We then tested the original segmentation of the genome into domains, using the Directionality Index method by Dixon et al[25] in mouse cortex Hi-C data (mean RMSE of 1.27). For each TAD, we estimated the optimal power-law parameters $a_i$ and intersect $b_i$ resulting with RMSE score of 1.20, an improvement of 7% compared to a random segmentation of the genome (using TAD shuffling, RMSE = 1.29). The hierarchical agglomeration of neighboring domains did not further improve the fit noticeably (RMSE = 1.19).

Finally, we considered a more sophisticated parametric family for modeling Hi-C interaction data in each TAD or merge area. As we noticed, many TADs do not follow a power-law distribution (straight line in log–log plots), but instead show a "broken" behavior, which could reflect one power-law fit for the closer distances, and another at more distant ones (Supplementary Fig. 3). For this, we developed a piece-wise power-law regression model for modeling the average number of interactions (in log scale) for any distance (in log scale) (Methods section). This richer model offers a much more accurate fit of the Hi-C data (RMSE = 1.06), a 12% reduction in fit error compared to the original power-law fit.

For comparison, RMSE for simulated data sampled (using Poisson distribution with matching "read depth") from the background model itself, was only 3% lower at RMSE = 1.03. Put together, hierarchical TAD models with bi-linear power-laws allow us to model Hi-C interaction data with high accuracy, thus forming a detailed background model against which we can compare the data and identify over-represented DNA–DNA interactions.

**Identification of enriched interactions in the mouse cortex.** We now wish to use the hierarchical TAD-specific bi-linear model as background model for Hi-C, and identify over-represented DNA–DNA interactions that could correspond to promoter–enhancer and other functional interactions in vivo.

For this, we aim to compute the "virtual 4C" plot for each promoter, and compare it to the expected number of interactions according to the background model. We consider a large genomic region surrounding each promoter (±1 Mb) and search for regions showing enriched Hi-C interactions with the promoter.

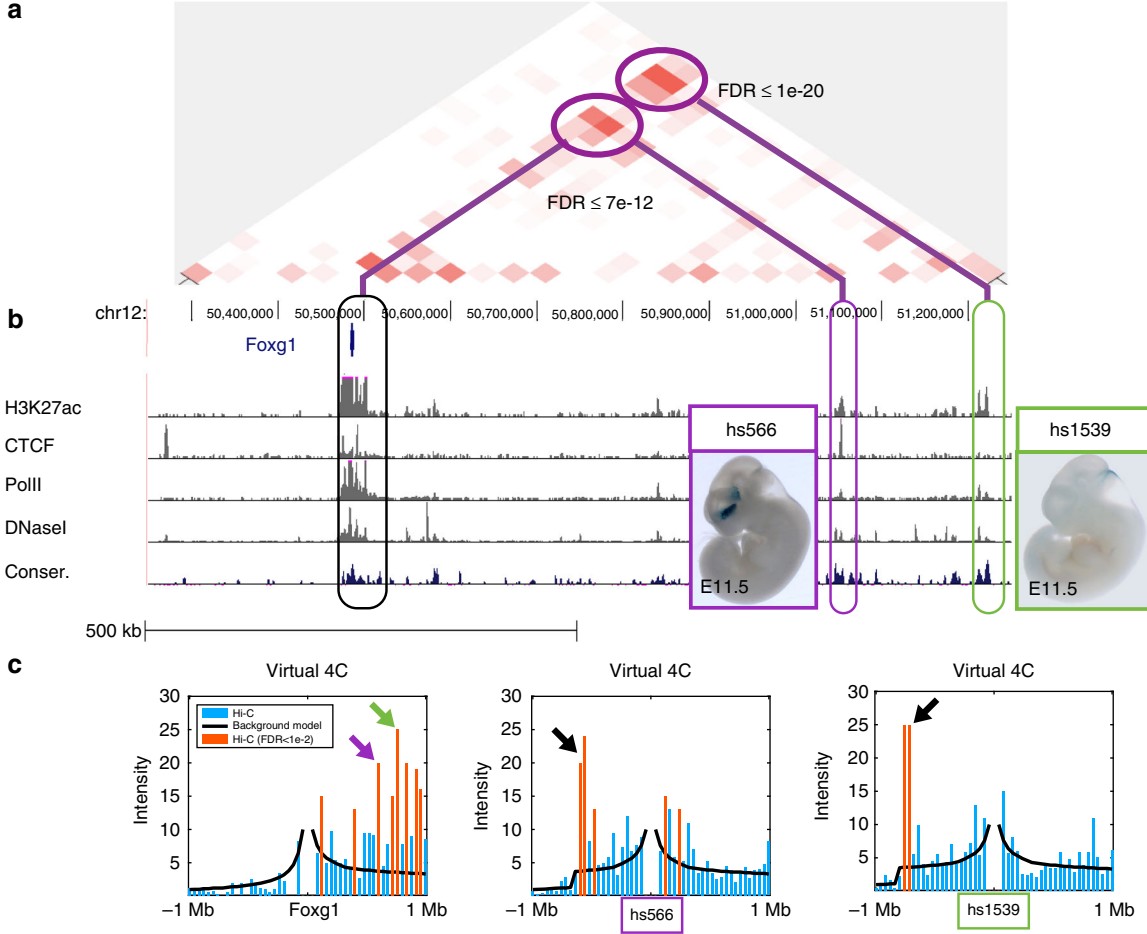

**Fig. 2** Analysis of mouse cortex Hi-C data by PSYCHIC. PSYCHIC analysis of the Foxg1 locus in adult mouse cortex Hi-C data[25] identifies two putative enhancer bins enriched with Foxg1. **a** Residual map for the Foxg1 locus (chr12, 50.3–51.2 Mb) shows the measured Hi-C map after the subtraction of the background model fitted by PSYCHIC, with two significantly enriched Hi-C cells, connecting Foxg1 with two putative enhancer bins. **b** ChIP-seq and evolutionary conservation data matching active enhancers, within the two putative enhancer regions. **c** Virtual 4C plots centered at Foxg1 (left) and the two enhancer loci (hs599 and hs1539), comparing measured Hi-C data (bars) vs. the fitted background model as reconstructed by PSYCHIC (black line). Statistically significant DNA–DNA interactions (FDR < 0.01) are marked by orange bars. Arrows show significant interactions between Foxg1, hs566 and the hs1539 orthologous regions. Inset images (hs566, hs1539) from the VISTA Enhancer Browser by Visel et al[55]

By subtracting the background model from the Hi-C data, we obtain the "residual" over-representation map. Statistical significance score (p values) are assigned using a log-Normal distribution fitted to the residuals in a 2 Mb window surrounding each promoter, then corrected for multiple hypotheses (FDR)[54] (Methods section).

We begin by focusing the Foxg1 locus (chr12, 50.3–51.2 Mb) using Hi-C data from mouse cortex[25]. Figure 2a shows the "residual" map for this locus. Prominent over-represented cells match two Foxg1 enhancers (hs566 and hs1539) located 550 Kb and 750 Kb downstream of the gene, with FDR values of 7e-12 and 1e-20, respectively. These two enhancers were discovered in human by us and others, using ChIP-seq and conservation data[55–57]. Comparison to published ChIP-seq data of H3K27ac, CTCF, PolII, and DNaseI hypersensitivity data from the mouse ENCODE project[58], and evolutionary conservation data[59] further identifies the exact location of these Foxg1 enhancers (Fig. 2b).

**Genome-wide validation of putative enhancers.** To further test our results on a genome-wide scale, we systematically characterized the chromatin landscape surrounding all predicted enhancers in mouse cortex[25]. For this, we aligned a 4 Mb region around each of the 17,788 putative enhancer regions (in Hi-C bin resolution) using an FDR threshold of 1e-2, and tested various enhancer-related chromatin marks. These include active enhancer and promoter marks (H3K27ac, H3K4me1, PolII), CTCF, evolutionary conservation, DNA accessibility, and chromHMM predictions[58–61] (Fig. 3, blue lines and heatmaps). For control, we also computed the average signal at a random set of genomic regions up to 1 Mb away from promoters (Fig. 3, dotted black lines). For all data types, the predicted enhancers were significantly enriched compared to their surrounding flanking regions (See Supplementary Fig. 4 for heatmaps of control regions).

Similar analysis for predicted boundaries identifies enrichment for CTCF and high DNA accessibility, as well as enrichment for promoter-like marks of PolII and H3K27ac, without H3K4me1 enrichment (Supplementary Fig. 5).

Next, we wished to study the effect different initialization methods have on the predicted promoter–enhancer interactions. For this, we initialized two-component intra- /inter-TAD model using three methods, including the Directionality Index[25], the Insulation Square method[23] as well as a random initialization of TADs. These changes had a limited effect on the predicted enhancer Hi-C bins (Supplementary Fig. 6).

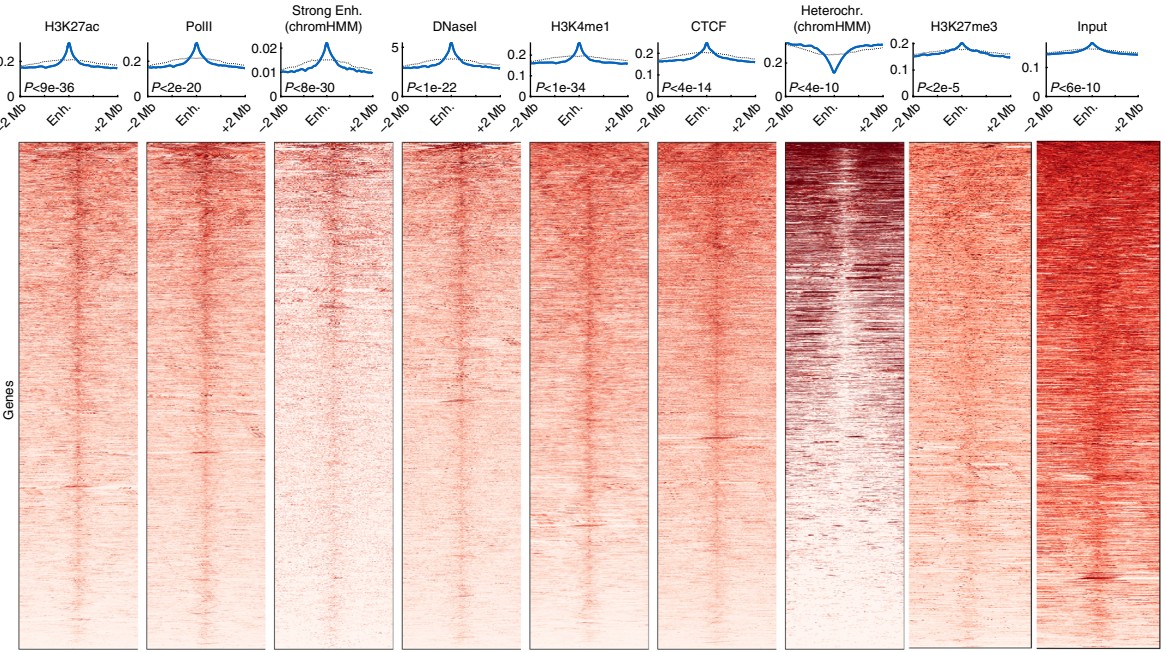

**Fig. 3** Chromatin marks surrounding predicted enhancer regions. Chromatin marks at 4 Mb windows centered around 17,788 putative enhancer regions, predicted using PSYCHIC (FDR < 1e-2) for adult mouse cortex Hi-C data[25]. Shown are typical enhancer marks (H3K27ac, H3K4me1), along with PolII and CTCF ChIP-seq, chromHMM classifications, H3K27me3, and DNaseI hypersensitivity assays. Blue lines mark the average signal over the predicted enhancer Hi-C bins. Black dotted lines mark the signal averaged over a random set of genomic loci, sampled in 2 Mb windows around promoters. The statistical significance of each plot (p value) is calculated by comparing the average signal at putative enhancers with their >500 Kb surrounding

We then turned to analyze the statistics of the predicted promoter–enhancer interactions. Overall, 49% of the predicted enhancers are located within 120 Kb of their target promoters, with only about 15% regulating the nearest gene (56% regulate one of the 5 nearest genes). About 87% of the predicted interactions fall within a topological domain (compared to 60% at random), and 92% comply are contained within the first hierarchical merge of TADs. Similar statistics were obtained to additional Hi-C data sets analyzed (see below) in human and mouse—overall, 88% of predicted enhancers are within the same TAD, compared to 45% in random shuffles (Fig. 4).

Next, we calculated the distribution over the number of putative enhancers regulating each gene, and compared it to the distribution of randomly selected regions (equivalent to a "random set" of near promoter loci). As shown in Supplementary Fig. 7, we observed a much greater number of genes predicted to be regulated by multiple enhancer regions, compared to the random set. Our results show some genes to be regulated by ten or more enhancers. For example, 443 genes are predicted to have five brain enhancer regions (FDR < 1e-2), compared to only two in the randomized set, or three expected according to a binomial distribution.

**A comprehensive catalog of human and mouse enhancers**. To obtain a comprehensive list of putative enhancer regions, we gathered Hi-C data in 15 conditions and cell types in human and mouse, including mouse cortex and embryonic stem cells[25], mouse embryonic stem cells, neural progenitor cells (NPC), and neurons[20], and mouse B-lymphoblast (CH12LX) cells[19], as well as human embryonic stem cells and lung fibroblast IMR-90 cells[25], GM12878 B-lymphoblastoid cells, and HMEC, HUVEC, IMR-90, K562, KBM7, and NHEK cells lines[19]. We then used PSYCHIC (with hierarchical TAD merging and bi-linear power-law fit) to identify over-represented interactions (up to 1 Mb) from promoter regions.

Globally, using an FDR threshold of 0.01, we predicted 267,938 putative enhancers (88,193 in mouse and 179,745 in human) that regulate a total of 25,783 genes (20,471 in mouse and 20,264 in human). A more stringent FDR threshold of 1e-4, yields 136,448 putative enhancer regions (38,405 and 98,043) regulating 21,435 genes (14,698 and 17,298 for mouse and human, respectively). These are summarized in Supplementary Table 1 (full lists in Supplementary Data 1, 2) or in our supplementary webpage www.cs.huji.ac.il/~tommy/PSYCHIC.

**Comparison to other algorithms for enriched interactions**. To test these predictions, we collected external ChIP-seq data in matching conditions, using which we can compare our predictions with their surrounding loci. In addition, we used previous sets of predicted DNA–DNA interactions for the same Hi-C data, by Fit-Hi-C[62]—that uses a chromosome-wide statistical model (with no TAD resolution) to identify enriched Hi-C cells—and HiCCUPS[19] —where the enrichment of each Hi-C cell is computed based on its neighboring cells. For an unbiased and systematic comparison, we identified all DNA–DNA interactions that involve promoter loci, predicted by HiCCUPS (for human IMR-90, GM12878, K562, HMEC, HUVEC and NHEK cell lines)[19], or Fit-Hi-C (human IMR-90 cells, and mouse cortex and ES cells)[62] and compared their ChIP-seq signal.

As shown in Fig. 5 and Supplementary Fig. 8, the predictions by PSYCHIC are generally more enriched (both in terms of absolute signal strength, and its genomic localization, or "sharpness") for H3K27ac, DNaseI, and chromHMM's "Strong Enhancer" class in matching cell types. We do observe, however, stronger enrichments for HiCCUPS' and Fit-Hi-C's predictions for both CTCF and chromHMM's "Insulator" loci, suggesting that these methods, that are not TAD-specific are possibly skewed by boundary elements, leading to over-estimation of near-boundary interactions (Supplementary Fig. 8).

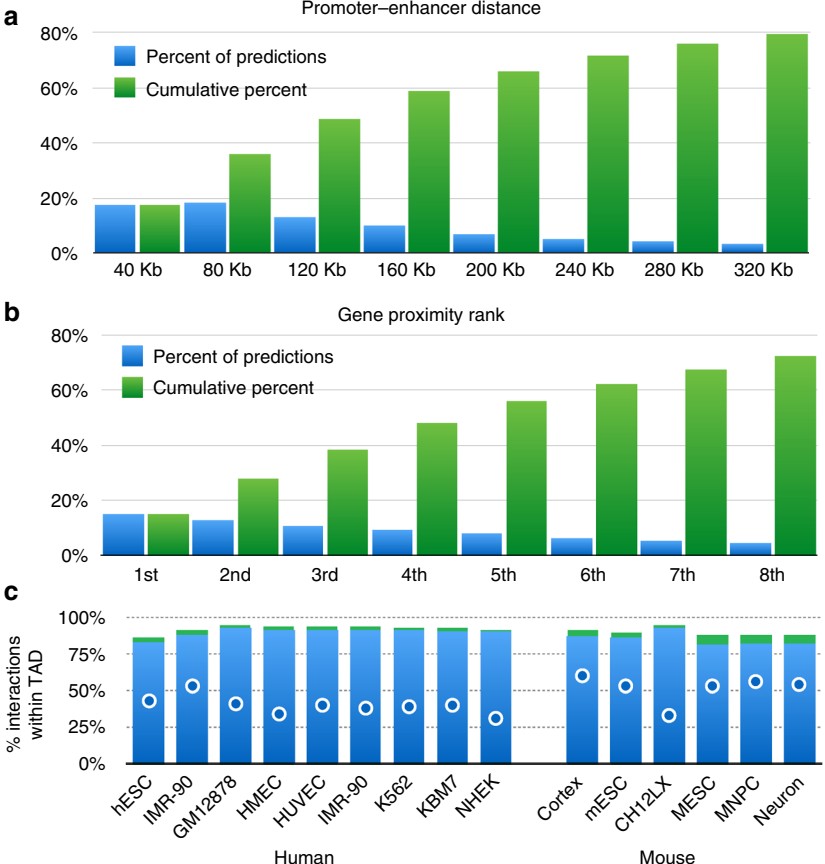

**Fig. 4** Promoter–enhancers interactions. **a**. Distribution (blue) and cumulative distribution (green) of promoter–enhancer interactions for mouse cortex data[25], as predicted form PSYCHIC (FDR < 1e-2). **b** Same as (A), reporting the proximity rank of genes associated with predicted enhancer bins. **c** Most putative enhancers reside within the same TAD as their targets. For each of the 15 human and mouse Hi-C experiments analyzed, the Y-axis shows the percent of predicted DNA–DNA pairs to fall within the same topological domain. Green supplements show the percent of additional pairs falling within 1st level of TAD–TAD hierarchical merges. Blue dots show percent of "random" enhancers residing within the same TAD

**Enrichment of eQTLs and nuclei cryo-sectioning**. To further test the quality of our predicted promoter–enhancer interactions, we computed their agreement with additional data sets. First, we analyzed the data from the Genotype-Tissue Expression (GTEx) Project (https://gtexportal.org), in which expression quantitative trait loci (eQTLs) were collected in multiple different human tissues by comparing the genotypes and expression level profiles in hundreds of donors[63]. As we show in Fig. 6a, the majority of our promoter–enhancer predictions are supported by GTEx eQTL data. These include, for example, 55% of our GM12878 predictions (at FDR < 1e-2) compared to only 20% of the random interactions, or 29–35% of HiCCUPS promoter–enhancer interactions. More stringent PSYCHIC thresholds further improve this data set agreement: 58% of 1e-4 predictions, or 63% of the predictions at FDR < 1e-10. Similar numbers are obtained for all other human data set analyzed. These numbers also outperform Fit-Hi-C predictions––for example, GTEx data support 25% of the human ESC Fit-Hi-C promoter–enhancer predictions (at q value < 1e-10) compared with 46% for our 2075 predictions (at FDR < 1e-2), or 29% for their 866 (at q < 1e-20) predictions compared with 48% for our 833 predicted interactions (at FDR < 1e-4).

In addition, we compared our prediction with DNA–DNA interactions in mouse ESC, predicted using ultra-thin cryo-sectioning slices through a single nucleus, followed by sequencing[64]. Here, we compared the average number of slices in which both the promoter and its predicted enhancer region are captured in the same slice. As shown in Fig. 6b, the 9771 promoter–enhancer interactions predicted by PSYCHIC for mouse ESC data (at FDR <

1e-2) are co-sequenced in an average of 41 slices ($p$ < 5e-92 using random shuffles), or 42 slices on average for the 3908 predictions at a threshold of 1e-4, compared to an average of 30 slices for random interactions, or 35 slices on average among the 7164 promoter–enhancer predictions of Fit-Hi-C (at a threshold of 1e-10). These results further support our methodology and the biological significance of our predicted enhancer regions and their associated target genes.

**Validation by capture Hi-C and ChIA-PET data**. Finally, we compared our promoter–enhancer interactions with other proximity-ligation data sets, including Capture Hi-C (CHi-C) data from mouse ES cells[21] and ChIA-PET data from GM12878 cells[22]. The Capture Hi-C interactions show high support for the predicted interactions by PSYCHIC, with coverage ranging from 69% of PSYCHIC predicted interactions (in mESC, called using an FDR threshold of 1e-2) to 74% (threshold of 1e-4), compared to 52–66% of Fit-Hi-C predictions for mESC Hi-C data (Supplementary Fig. 9a). Next, we compared our predictions to ChIA-PET data in GM12878 cells[22]. ChIA-PET interactions obtained using PolII antibodies showed high support for our promoter–enhancer predictions, covering 37% (PSYCHIC GM12878 predictions with threshold of 1e-2) to 55% (threshold of 1e-10); compared to 33–36% for HiCCUPS GM12878 calls (Supplementary Fig. 9b). Intriguingly, a higher portion of HiC-CUPS calls (73%) was supported by the ChIA-PET data using CTCF antibodies, compared to ~34% for PSYCHIC. This is in

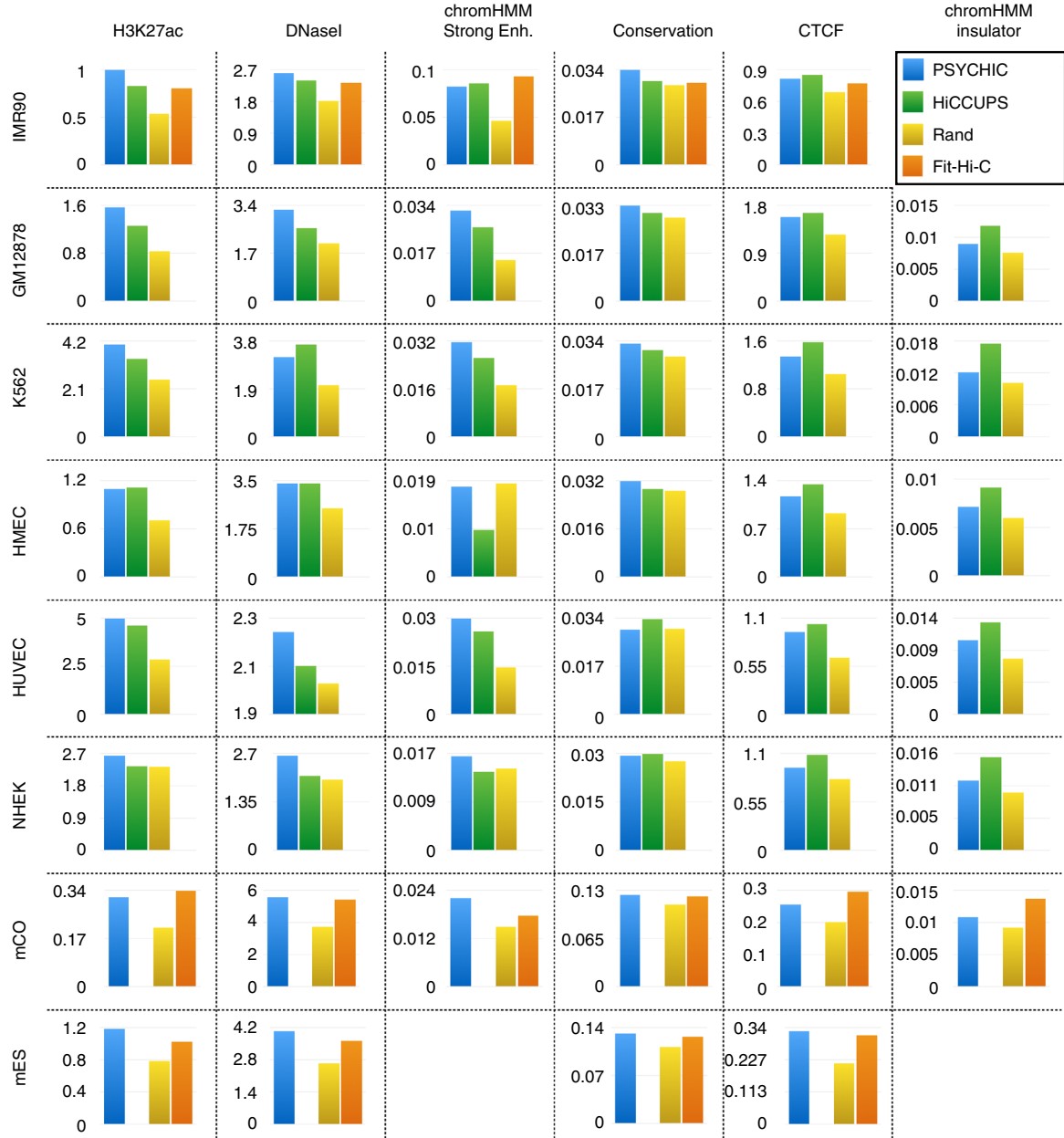

Average ChIP-seq signal for predicted enhancer-promoter interactions

**Fig. 5** Promoter–enhancer predictions are supported by ChIP-seq data, DNA accessibility, evolutionary conservation, and chromHMM "Strong Enhancer" loci. Shown are the average signal for ChIP-seq and additional genomic data, over predicted enhancer regions using PSYCHIC (FDR < 1e-2, blue), HiCCUPS (green), Fit-Hi-C (q value < 1e-10 for IMR90 and hES, q < 1e-4 for mCO, orange) or random interactions (yellow). Notably, PSYCHIC predictions are generally more enriched for all enhancer-related data, while HiCCUPS and Fit-Hi-C predictions are more enriched for CTCF and Insulator marks. No chromHMM data was found for mES

line with the relative enrichment of CTCF ChIP-seq signal among HiCCUPS predictions (Fig. 5).

**Interaction with inactive enhancers**. Notably, most—but not all—putative enhancer regions show strong enrichment for active chromatin marks. For example, ~70% of the enhancers predicted with FDR < 1e-2 show increased accessibility compared to their flanking DNA regions (Fig. 3, "DNaseI"). Almost half (46%) of predicted enhancer regions show enrichment that is greater than one standard deviation compared to their flanking regions (32% > 2 SD). For comparison, only 43% of the randomly selected regions show increased accessibility, with only 24% exceeding one

standard deviation (15% > 2 SD). Similar numbers are obtained for H3K27ac or CTCF.

This suggests that over-represented DNA–DNA interactions (in Hi-C) are not limited to active and accessible regions, and raises the hypothesis that a non-trivial fraction of putative enhancer regions are "silent" and inaccessible. A closer examination identified several known enhancers even within those. For example, PSYCHIC identified the ZRS locus as interacting with the *Shh* gene, even in adult mouse cortex (Fig. 7). In the mouse, early developmental *Shh* expression is essential for autopod formation, regulated in developing limbs by the distal ZRS enhancer, located ~1 Mb away[8,65]. Our results suggest that ZRS is in close physical proximity to *Shh* even in adult brain. Analysis of

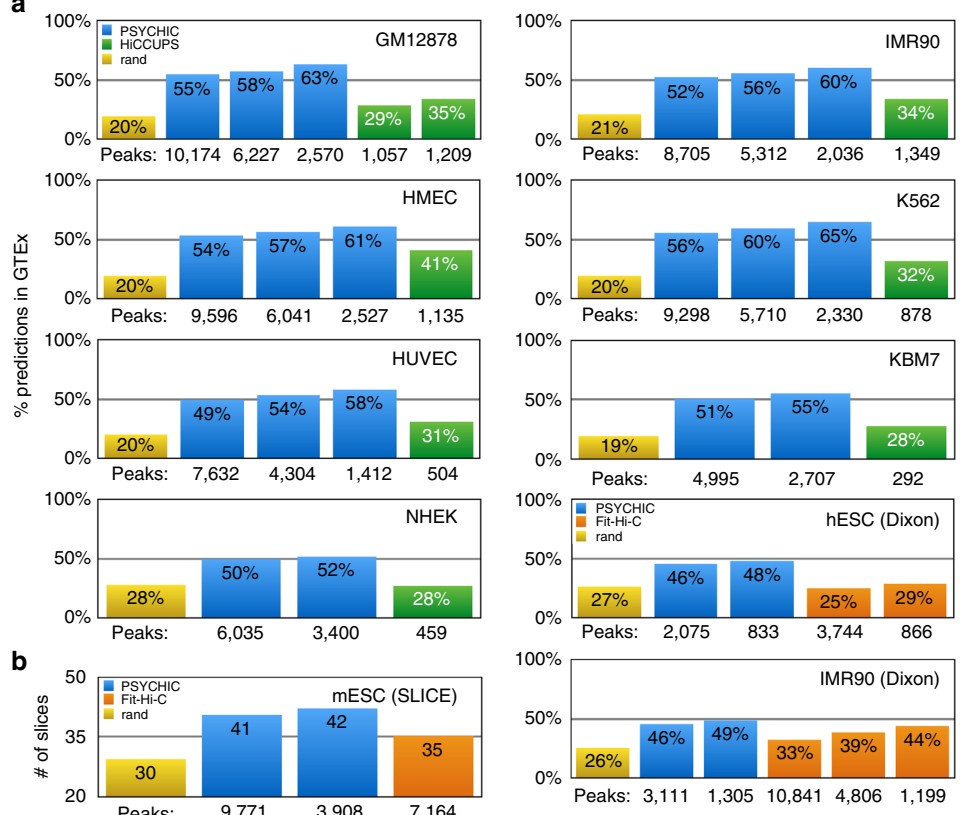

**Fig. 6** PSYCHIC predictions are enriched for eQTLs and ultra-thin nuclear cryo-sectioning slices. **a** The majority of PSYCHIC's promoter–enhancer interactions are supported by eQTL data from the Genotype-Tissue Expression (GTEx) Project[63]. Shown is a comparison of the percentage of predicted interactions using various methods with eQTL data, including random promoter-proximal interactions (yellow), PSYCHIC predictions (blue, using FDR thresholds of 1e-2 and 1e-4. Results with FDR < 1e-10 are also shown for first 5 cell lines), HiCCUPS[19] (green), or Fit-Hi-C[62] (orange; q value thresholds of 1e-10, 1e-20, and for IMR90 also 1e-40), for various cell lines and tissues in human and mouse. Below each bar, we mark the number of predicted promoter–enhancer interactions. **b** Comparison of random (yellow), PSYCHIC (blue; FDR < 1e-2 and FDR < 1e-4) and Fit-Hi-C[62] (orange; q value <1e-20) predictions, with regard to the sequencing of genomic DNA content in ultra-thin cryo-sectioning slices in mES cells[64]. Y-axis marks the number of slices in which both the enhancer and the promoter regions were co-sequenced

Hi-C data in mouse and human identifies similar interactions between *Shh* and ZRS in most mouse conditions (Supplementary Fig. 10a). This was recently validated by DNA FISH showing ZRS in the proximity of *Shh* throughout a variety of tissues and developmental stages, while not being in active transcription[66]. Similarly, a cross-condition analysis of the promoter–enhancer interactions (predicted using PSYCHIC, GM12878, with a stringent threshold of FDR < 1e-10) shows that >25% of these putative interactions are predicted (by PSYCHIC) in at least three additional human Hi-C data sets (compared to only 3% in random; Supplementary Fig. 10b).

## Discussion

In this work we presented PSYCHIC, a computational model for analyzing the Hi-C data to identify enriched DNA–DNA interactions. Using a probabilistic model and efficient algorithms, PSYCHIC identifies the optimal segmentation of chromosomes into topological domains, assembles them into hierarchical structures, and fits a TAD-specific background model for the Hi-C data. By considering a "virtual 4C" plot for every gene, and using the background model for statistical assessments, our algorithm identified 267,938 significant over-represented enhancer–promoter interactions in 15 Hi-C experiments in human and mouse.

To segment the genome into TADs, our algorithm uses a probabilistic two-component model that independently computes for every cell in the Hi-C matrix, the likelihood ratio between intra-TAD and inter-TAD models. This score assigns similar importance to near and far DNA–DNA interactions, and is less affected by short-range interactions that dominate Hi-C data, but are mostly invariant of topological domains. This additive score is easily computed from nested TADs, allowing for fast and scalable Dynamic Programming algorithm.

Our algorithm then computes for each TAD the average number of contacts at any distance. This spectrum was previously modeled using power-laws, which we replaced by two-segment models, greatly improving the model accuracy. These results suggest a transition between two packaging mechanisms, typically at 100–300 Kb.

Currently, most Hi-C data are of 10–40 Kb resolution, hindering our ability to pinpoint promoter–enhancer interactions. Various methods (e.g., ChIP-seq, accessibility, evolutionary conservation) could be applied to further identify enhancers in higher resolution. As more detailed Hi-C data are accumulated, PSYCHIC will offer more accurate predictions. While the running time of PSYCHIC is quadratic, it is scalable. Various heuristic assumptions (e.g., maximal size for sub-TADs) will dramatically speed it up, allowing for higher resolution analysis using future Hi-C data sets.

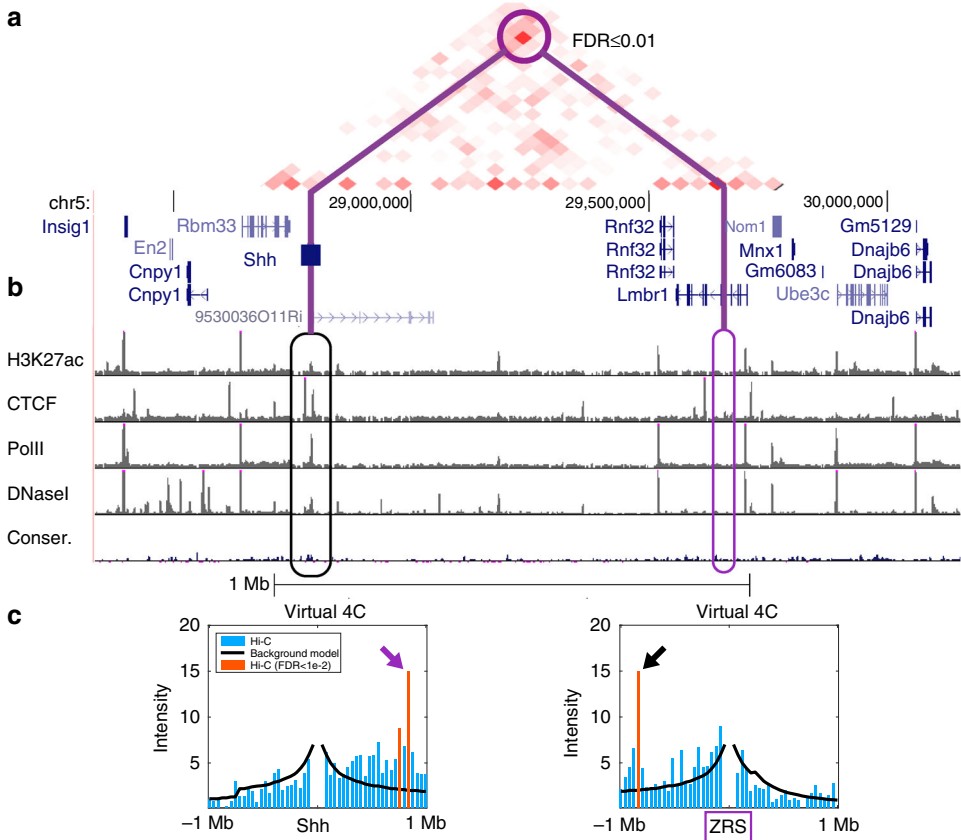

**Fig. 7** Shh–ZRS interaction in adult mouse cortex. Significantly enriched promoter–enhancer interactions (in adult mouse cortex) between *Shh* and inactive limb-specific enhancer ZRS (chr5:28.3–30.2 Mb). **a** Residual map (measured Hi-C data after subtraction of background model fitted by PSYCHIC) identifies over-represented DNA–DNA interactions between the *Shh* locus and its limb-specific enhancer ZRS. **b** Genome-wide ChIP-seq and accessibility data from adult mouse cortex show no active enhancer marks for this enhancer. **c**. Virtual 4C plots for the *Shh* (left) and the ZRS (right) loci, comparing Hi-C interactions with the local background model reconstructed by PSYCHIC. Arrows mark significant between *Shh* and ZRS

Ground-truth data for promoter–enhancer interactions are still limited, and we have taken multiple approaches to establish our predictions. We showed that the predicted enhancer regions are enriched for active marks (H3K27ac, H3K4me1, PolII), DNA accessibility, or CTCF. This was shown initially for a single locus (*Foxg1*) in the mouse cortex, and later supported in a genome-wide manner over multiple tissues. Comparison to previous methods, including HiCCUPS and Fit-Hi-C, generally showed stronger and sharper enrichment for PSYCHIC, as well as a general bias of other algorithms to near-boundary interactions. Secondly, we used high-throughput eQTL data, linking genotypes and gene expression profiles in hundreds of donors, and intersected them with our predictions. As we show, about half of PSYCHIC's predictions are supported, in a variety of cell types. Finally, we used recently published cryo-sections of nuclei, showing that predicted promoter–enhancer pairs are co-sliced more often then expected.

Intriguingly, a closer examination reveals that ~1/3 of predicted regions are inaccessible and bear no active chromatin marks. These include the ZRS locus that acts as a limb-specific distal enhancer for *Shh*, located nearly ~1 Mb away. While the ZRS locus shows no accessibility or ChIP peaks in the mouse cortex, therefore predicted to be inactive, it presents a significant number of interactions with *Shh*. Indeed, Williamson et al.[66] recently used FISH and 5C to show that ZRS and *Shh* are located in spatial proximity regardless of their activity.

These results suggest that the 3D structure of the genome may be organized to support regulatory DNA–DNA interactions, rather than merely reflect the set of accessible or active regions in the genome. As more Hi-C data are collected and analyzed, we hope to shed light on the causality of gene regulation and genome packaging, as well as the plasticity of genome packaging in general.

Put together, we demonstrated how Hi-C data—typically used to identify TAD boundaries—can be used to identify enriched DNA–DNA interactions, including thousands of putative enhancer regions and associate them to their target genes.

## Methods

**Modeling Hi-C data**. Intra-TAD Hi-C data are represented using log-Normal distribution with two parameters (mean and standard deviation) for each distance $d$

$$P_d(N|\text{intra}) = \log\text{–Normal}(\mu_d^{\text{intra}}, \sigma_d^{\text{intra}}) \tag{9}$$

where the log-Normal distribution with mean $\mu$ and standard deviation $\sigma$ can be written as:

$$P(x) = \frac{1}{x\sigma\sqrt{2\pi}} e^{-(\log x - \mu)^2 / 2\sigma^2} \tag{10}$$

Inter-TAD Hi-C data are represented similarly:

$$P_d(N|\text{inter}) = \log\text{–Normal}(\mu_d^{\text{inter}}, \sigma_d^{\text{inter}}) \tag{11}$$

Bayes' law could be used to derive the posterior probabilities of the intra-TAD:

$$P_d(\text{TAD}|N) = \frac{P_d(\text{TAD})}{P_d(N)} \times P_d(N|\text{TAD}) \tag{12}$$

and inter-TAD models:

$$P_d(\text{BG}|N) = \frac{P_d(\text{BG})}{P_d(N)} \times P_d(N|\text{BG}) \tag{13}$$

given the number of interactions $N$ at a given distance $d$, and the prior probabilities $P_d$(intra) and $P_d$(intra).

**Bi-linear regression of log-intensity and log-distance**. We model the Hi-C interaction intensity between two loci as a segmented power-law function of their distance. In log–log scale this is modeled by a two-piece segmented linear regression model. For this, we developed a computational algorithm (implemented in MATLAB) to iterate over the optimal breaking point and estimates the two parameters (intercept and slope) for each segment, while minimizing the squared deviation of the data (in log–log scale). Similarly, a piece-wise linear model was learned for the remaining inter-TAD regions ("Sky").

**TAD merges**. Neighboring TADs are merged into a hierarchical structure, according to a "merge score" that compares the mean Hi-C intensity per distance within the two underlying TADs, their inter-TAD area, and the null inter-TAD model (represented by $\alpha$ in Eq. 10). We then iteratively merge two neighboring TADs whose merge area is the most similar, up to a maximal domain size of 5 Mb.

**Random set of enhancers**. A random set of genomic loci along the genome, while maintaining a similar distribution around gene promoters, we considered for each gene all genomic loci up to 1 Mb away (on either direction), and selected each with a probability of 1e-2.

**Statistical significance of ChIP-seq for putative enhancers**. To estimate the statistical significance for the average ChIP-seq signal (or others) at putative enhancer regions (Fig. 3), we fitted a Normal distribution to the average ChIP-seq signals at distances >500 Kb from the predicted enhancers, then approximated the $p$ value as the cumulative distribution function (CDF) given by the Normal distribution at the average ChIP-seq signal for predicted enhancer regions.

**Simulated Hi-C data**. Hi-C matrices were simulated by sampling considering the hierarchical TAD-specific fit model (from PSYCHIC), then re-sampling each Hi-C cell from a Poisson distributions with a parameter $\lambda$ matching the expected mean number of DNA–DNA interactions.

**Statistical enrichment score**. To assign a statistical significance score ($p$ value) for each putative enhancer (namely, an over-represented interaction between a promoter region and some other locus), we assumed a Normal distribution of the local residual map (i.e. Hi-C minus PSYCHIC background mode) at a 2 Mb surrounding the promoter of each gene. We then fitted maximum likelihood estimator for the mean value $\mu_i$, and its standard deviation $\sigma_i$, and used these statistics to translate the deviation of each Hi-C cell from its background model, into $z$-scores. Finally, we assigned a $p$ value for each $z$-score using a standard Normal cumulative distribution function, and applied an FDR correction for multiple hypotheses[54].

**Hi-C data sources and preprocessing**. Normalized Hi-C maps were analyzed. For Dixon et al[25], normalized Hi-C data at 40 Kb resolution were obtained from the Ren lab website (http://chromosome.sdsc.edu/mouse/hi-c). For Rao et al[19], processed data (intra-chromosomal, MAPQGE30, KR normalized) were downloaded from GEO (GSE63525), and down-sampled from 5 Kb to 25 Kb resolution for higher coverage and more robust analysis. For Fraser et al[20], processed and normalized Hi-C data were downloaded from GEO (GSE59027) in 50 Kb or 100Kb resolution.

**Statistical significance of SLICE data**. To quantify the statistical significance of the average number of promoter–enhancer co-occurrence in the cryo-sectioning slices, we randomized our predictions 1000 times by shuffling the gene names (stratified by chromosomes). We then computed the average slice co-occurrence in each shuffle. PSYCHIC predictions outperformed all 1000 shuffles, and obtained a Normal distribution $p$ value of 5e-92.

**Code availability**. PSYCHIC is publicly available via GitHub (https://github.com/dhkron/PSYCHIC).

**Data availability**. A full list of putative enhancer regions, as well as the genes they regulate is available in Supplementary Table 1 and Supplementary Data 1, 2, and in our supplemental website at www.cs.huji.ac.il/~tommy/PSYCHIC. Also available in our website are saved UCSC Genome Browser sessions for mouse (mm9) and human (hg19).

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

## Acknowledgements

We would like to thank Nir Friedman, Eran Rosenthal, Shira Strauss, and members of the Kaplan lab for helpful discussions and comments. T.K. is a member of the Israeli Center of Excellence (I-CORE) for Gene Regulation in Complex Human Disease (no. 41/11) and the Israeli Center of Excellence (I-CORE) for Chromatin and RNA in Gene Regulation (no. 1796/12). This research was also supported by a Marie Curie Career Integration Grant (PCIG13-GA-2013-618327), and an Israel Science Foundation grant (no. 913/15) to T.K. Y.G. is supported by a Leibniz Fellowship.

## Author contributions

Conceived and designed the method: G.R. and T.K. Implementation: G.R. Analyzed the data: G.R., Y.G., D.M., and T.K. Wrote the paper: G.R. and T.K.

## Additional information

**Competing interests:** The authors declare no competing financial interests.

