## [Peer Review File · Nature Communications]

Reviewers' comments:

Reviewer #1 (Remarks to the Author):

In this article the authors present a new computational method to analyse HiC retrieving an hierarchy of TADs and inter-TAD domains. Their two-component model also gives the ideal background estimation to detect over-represented chromatin interactions.

The work is exciting and timely due to the high demand in the field for new methods bridging HiC data (typically low res., 10s Kb) with the finer scale needed to specify promoter-enhancer interactions. The approach is sound and well explained but the details given are unbalanced and important controls and results are missing. I recommend major revision of the article.

Major points:

1. While it is important to explain the method proposed, many of the details given are ill-suited for a results section (e.g. how log-normal distribution is written and many standard bayesian eqs. presented should change to methods or appendix). On the other hand an important output of the proposed method is TAD boundaries and the inter-TAD structure captured but these are not really shown or discussed. A comparison with DI, contrast index or insulation square would be good to see. In line 152 the authors state they obtain a "forest like TAD structure" but this cannot be seen which makes the model abstract to the reader.
2. The chromatin enrichments in Fig.3 used to test their predictions are interesting but given that a third are negative I wonder what is the value of the analysis shown. Also it needs to be supplemented with a mock chip/ input. Why was chromHMM used and not directly heterochromatin marks ? Depletion of features is also interesting to see here.
3. A hallmark of enhancer marks is their increased tissue specificity and here a big opportunity is being missed. I wonder if the authors' method could find tissue-specific enhancers from the multiple HiC data analysed or if invariable contacts like the SHH-ZRS shown are predominant. Shh has multiple enhancers controlling its expression in different tissues and developmental times - do the virtual 4C plots change in other data considered?
4. The analysis is centred on a 2Mb region around promoters. Why not unbiasedly genome-wide to find potential enhancers even further away? The sentence on putative enhancers and their potential target genes should be made clearer (lines 218-220). How do the authors know the target promoter and what are the assumptions (proximity and single promoter?) ?

Minor points:

1. Inter-TAD merging was restricted to 5Mb but no justification given. Additionally it is a bit confusing that the paper considers inter-TAD interactions but BG is used in equations. Are these background or meaningful interactions?
2. Using average DNA-DNA interaction as summary statistic for each domain can also lead to biases - not clear why the authors consider this better.
3. The two-component model is interesting but the justification given for using it should be better supported and explored. I think the reader would benefit from seeing these data plotted. Often there were Fig. S1 references where the plots did not correspond to expectations.

Reviewer #2 (Remarks to the Author):

Promoter-Enhancer Interactions Identified from Hi-C Data using Probabilistic Models and

Hierarchical Topological Domains, by Ron et al.

This manuscript describes a computational approach, 'PSYCHIC', to analyse Hi-C data and identify topological domains and putative enhancer-promoter interactions within them. The method takes an input set of topological domains, refines them based on a probabilistic model to predict whether the interaction between a pair of bins is most likely to be within or between TADs, and then iteratively merges pairs of adjacent domains to produce a hierarchical TAD structure. The authors use this hierarchical TAD model to inform modeling of the expected number of interactions for any distance, within or between TADs, and to identify enriched interactions of promoters with other bins. Using this method, the authors are able to identify interactions with known enhancers for *Foxg1* and *Shh*. They further identify putative enhancer-promoter interactions using multiple Hi-C datasets from human and mouse, and show that putative enhancer bins are enriched for eQTLs and other features previously associated with enhancer activity.

Taking into account local background, such as whether an interaction occurs within or between topological domains, is crucial when determining significance of interactions. The authors have provided a useful tool for determining significant interactions while identifying and taking into account topological domain structures. Some additional comparisons with existing datasets would be useful to further characterize the performance and robustness of the method. In addition, there are some improvements that could be made to the software provided that would increase its usability.

Major points

- How much does the performance of the algorithm depend on the initial set of TADs, i.e. how robust is the method? The authors could address this point by showing how the final TAD calls are affected by using random TAD positions or a different initial TAD-calling algorithm.
- The manuscript doesn't state what binning resolution is used for any of the Hi-C datasets. This should be included.
- The methods should also include details of the Hi-C processing and format of input data.
- Figure 3: 4Mb windows around putative enhancers are far too large to assess enrichment of chromatin marks at the enhancers – this is more reflective of the enhancer bins being in active vs inactive compartments. This figure should be replaced with a higher resolution analysis to support the statement that the enhancer regions are enriched in active chromatin marks.
- It would be informative to know how well these enhancer-promoter interactions overlap with interactions determined using other types of data, such as ChIA-PET (e.g. PMID 26686651) or promoter capture Hi-C (e.g. PMID 25752748).
- The authors report that detected enhancer-promoter interactions co-occur in a higher number of GAM slices than random interactions. What is the statistical significance of this increased co-occurrence?
- There is not much discussion of the features of the hierarchical TAD structure. Some examples of this structure at specific genomic loci would help readers understand the impact of the hierarchical merging. It would also be useful to include a comparison of this hierarchical approach with existing tools such as *Armatu*s (PMID: 24868242).
- Software: I was unable to fully test the software, since I don't currently have access to a system with Matlab installed. However, I downloaded and ran the software on a system without Matlab installed, where it unexpectedly ran without producing any errors other than empty files with the results of the calculations. The usability of the software tool would be greatly increased by adding a few extra features:
 - o Checks for required software at the start of the run, with error messages/installation directions if necessary
 - o Usage/help information available using "--help" or "-h" or similar
 - o A description of the expected output files and their format
 - o A description of the required format for input files
 - o List required versions of all prerequisite software

Minor points

- The authors refer to the bins detected as having enriched interactions with promoters as “enhancer regions” – a more neutral term that acknowledges the limited resolution of the Hi-C data would be more appropriate, such as “interacting bins” or “putative enhancer bins”.
- There are a few missing words / typos in the manuscript, e.g. line 12 “Proximity ligation methods such as Hi-C”, line 255 “are possibly skewed by boundary elements

Reviewer #3 (Remarks to the Author):

Ron et al present a new method to analyze a Hi-C (chromatin interaction) data set to reveal DNA-DNA interactions and then, by examining high confidence interactions with a promoter, they predict putative enhancers likely to regulate a gene. They provide circumstantial evidence in favor of the quality of the predicted enhancer collection, by showing enrichments of various enhancer-related properties. They also show that various enrichments are better than those seen when using state of the art methods for analyzing the same data to the same end (predicting enhancers). The paper is very well written.

Specific comments:

The finding shown in the last subsection (Figure 7) is exciting and intriguing. To me the statement “the hypothesis that a non-trivial fraction of the putative enhancer regions we have identified are “silent” and inaccessible. A closer examination identified several known enhancers even within those” was the most interesting aspect of the entire paper, and I was wishing that the authors had made the point more forcefully (e.g., by at least providing some basic numbers and listing more examples like ZRS-Shh in the supplement or even in main text), to make readers think of the possibilities here, even if there is the usual risk of false positives.

To what extent is the difference in predicted enhancers between the new method and previous methods due to difference in segmentation versus difference in how intra and inter-TAD interaction intensities are normalized to predict significant interactions? I am guess that the segmentation may not be very different and that the main differences seen in Figs 5 and 6 arise from the normalization procedure.

The idea of subtracting the expectation can be a double edged sword in this application. In comparing the residuals of different windows (when constructing the Normal distribution), the assumption is that enhancers may be present at all distances with equal prior probability. If in reality there is a much greater crowding of enhancers closer to (say within 100 Kb of) the gene, then the Hi-C data will persistently show strong intensities and the background model might discount that stronger signal. I am not sure what it does to the signal/noise ratio, but it clearly hurts the sensitivity of the method, in a way biased against closer enhancers.

The score $S(t)$ is somewhat intriguing since it is a heuristic as far as I can tell. If the “outside of TAD t ” pairs always involve going a fixed distance (h) outside of the TAD, then the number of such “outside” pairs scales linearly with the TAD length and the number of inside pairs scales quadratically with TAD length. What effect does this have on $S(t)$ and are these values comparable across different domains ?

The score ‘Score(C)’ (equation 8), in adding $S(t)$ over multiple (non-overlapping) TADs appears to

include the negative part of equation(7) RHS potentially multiple times. Am I missing something?

The motivation and practical utility of the hierarchical structure formation is not entirely clear. While it is clear that visualization of the HiC maps suggests the presence of such structures, how exactly do the authors propose the user of this tool will utilize the hierarchical structure?

Suppl Figure 2 needs error bars, possibly based on bootstrap sampling, for each bar. (No need for p-values, standard errors for each estimate will be just fine.)

I am not entirely clear why the fourth from left bar in Supplementary Figure 2 is called 'Hierarchical TADs'. I saw that the description of the model talks about hierarchical agglomeration of neighboring domains, followed by local parameter estimation. What is not clear is why the agglomeration had to be done to assign TAD (and inter-TAD) specific parameters for modeling local intensity. Couldn't the same exact result be obtained just from the knowledge of all TADs (the primary segmentation) followed by local parameter assignments? Or is there some sort of threshold applied to the hierarchical organization that determines which inter-TADs will be considered and which not. In short, I am not clear on why the hierarchical structure was needed in this step.

How different would the putative enhancer set be if using the simple power law rule for computing residuals in each promoter's 4 Mb context, followed by the same z-score / p-value calculation, but without utilizing the hierarchical structure, and even the TAD segmentation? I would expect that the difference is likely not major for within-TAD interactions, and any significant differences are probably in inter-TAD regions, which hosts a small minority (~10%) of putative enhancers anyways.

Figure 5 provides evidence that the predicted putative enhancers might be more reliable than those predicted by other methods on the same Hi-C data. However, the height of each bar (in each panel) is the 'average signal' of a certain type in the corresponding collection of predicted enhancers, and leaves open the possibility that there is a precision-recall tradeoff in play here, and that the average signal comparisons might look different at alternative values of cutoffs used for the other methods. I noticed that the authors have tried to keep this in mind in the evaluations of Figure 6A, where they show higher precision even with greater numbers of predictions made by their method.

While the authors have chosen to validate their predictions using various other sources of enhancer-related information, and while these are helpful in making their point, these do not quite reach the level of 'validation'. I would have been much more convinced if the authors had pointed out, for instance, that one or both of the 'true' Foxg1 enhancers would not have been picked by a serious scientist who did not have access to the method proposed here. A more systematic evaluation might be based on enhancer databases such as VISTA enhancer browser.

The finding shown in the last subsection (Figure 7) is exciting and intriguing. To me the statement "the hypothesis that a non-trivial fraction of the putative enhancer regions we have identified are "silent" and inaccessible. A closer examination identified several known enhancers even within those" was the most interesting aspect of the entire paper, and I was left wishing that the authors had made the point more forcefully (e.g., by at least providing some basic numbers and listing more examples like ZRS-Shh in the supplement or even in main text), to make readers think of the possibilities here, even if there is the usual risk of false positives.

Minor comments/ typos:

Line124: "all inter-TAD outside of TAD t, defined by pairs <i,j> " This needs some rewording/correction. You mean "all inter-TAD pairs outside of TAD t, defined by pairs <k,l>" ?

Line 141: "Specifically, we wish to iteratively agglomerative neighboring" Change 'agglomerative' to 'agglomerate' ?

Line 169: "significance over-represented interactions" Missing "of" ?

Line 155: "Once we have segmented the Hi-C map into hierarchical domains, we wish to model the expected intensity of the Hi-C map". At this point, the authors should provide a brief motivation of where they want to go with this. Very related point: I was for some time confused by the intent of Supplementary Figure 2, because it seemed to show that more parameterized models give better fits to data, which is completely expected. The authors put my doubts to rest with their last sentence in the subsection, that the goal is to build a more accurate background model to assess deviations from. This is of course a perfectly reasonable strategy and the authors should note it up front.

Reviewer #1 (Remarks to the Author):

The work is exciting and timely due to the high demand in the field for new methods bridging HiC data (typically low res., 10s Kb) with the finer scale needed to specify promoter-enhancer interactions. The approach is sound and well explained but the details given are unbalanced and important controls and results are missing. I recommend major revision of the article.

First, we would like to thank Reviewer 1 for the thorough and constructive review.

Major points:

1. *While it is important to explain the method proposed, many of the details given are ill-suited for a results section (e.g. how log-normal distribution is written and many standard bayesian eqs. presented should change to methods or appendix).*

The technical details have been transferred to the Methods section.

On the other hand an important output of the proposed method is TAD boundaries and the inter-TAD structure captured but these are not really shown or discussed. A comparison with DI, contrast index or insulation square would be good to see.

As we have showed in Supplementary Figure 2 (now updated to cover all mouse chromosomes) PSYCHIC's TAD calling allows for a more accurate log-Normal modeling of the Hi-C data, compared to Dixon et al TAD calling (DI method). We have also fixed a minor bug in the Y-axis scaling.

In addition we have analyzed our predicted TAD boundaries (currently Supplementary Fig 5), which showed enrichment of CTCF, PolII and accessibility.

Yet, the main scope of the PSYCHIC algorithm is to identify promoter-enhancer interactions. A better segmentation into TAD, together with the inter-TAD hierarchical structure are mostly a means for a better fit of the Hi-C data, thus allowing for a better identification of promoter-enhancer interactions. As we have showed in Figure S2 (now updated to cover all mouse chromosomes) PSYCHIC's TAD calling allows for a more accurate log-Normal modeling of the Hi-C data, compared to Dixon et al TAD calling (DI method).

Following the reviewer request, we have now initialized the two-component models with additional methods to the Directionality Index (Dixon et al, used by PSYCHIC initially). These include Insulation Score (using a running square, Crane et al, Nature 2015) as well as random initialization of TADs. These changes had a limited effect on the predicted enhancer regions (using a p-value threshold of $1e-4$) on adult cortex Hi-C data from mouse chr10 (Dixon et al). These results are now shown as Supplementary Figure 6:

In line 152 the authors state they obtain a "forest like TAD structure" but this cannot be seen which makes the model abstract to the reader.

We apologize for the unclear phrasing. By "forest-like TAD structure" we meant that TADs are only merged up to certain size (5 Mb), so the overall model is composed of multiple tree-like structures, as shown in Figure 1. We have now clarified this sentence.

2. The chromatin enrichments in Fig.3 used to test their predictions are interesting but given that a third are negative I wonder what is the value of the analysis shown. Also it needs to be supplemented with a mock chip/ input. Why was chromHMM used and not directly heterochromatin marks? Depletion of features is also interesting to see here.

We have initially included ChIP-seq data collected by Dixon et al, from adult mouse cortex. Indeed, some of the putative enhancer regions show no enrichment compared to their 4Mb surrounding regions. While this could be due to false positive calls, it has been shown that inactive enhancers could be colocalized by their target genes (e.g. for Shh-ZRS interactions, Williamson et al, 2016).

We have now included in Figure 3, following the reviewer's request, additional plots of H3K27me3 (from embryonic mouse brain, Ren lab, ENCODE) and Control (Ren Lab / ENCODE, adult cortex). Both showing only a marginal enrichment at putative enhancer regions (possibly explained by the high accessibility of these regions). We have also included a statistical analysis of these plots (by comparing the center point with distribution of average ChIP-seq values at distances >500Kb).

ChromHMM signal was used as a proxy, summarizing the chromatin state (for various active and repressive marks), which is present in multiple cell types (see Supplementary Figure 8). We have now also included matching heatmaps for the random regions (randomly chosen up to 2Mb away from each TSS, see Supplementary Figure 4). None of these plots were found to be statistically significant, except for CTCF (p-value<0.01), reflecting some enrichment of CTCF near gene promoters.

3. A hallmark of enhancer marks is their increased tissue specificity and here a big opportunity is being missed. I wonder if the authors' method could find tissue-specific enhancers from the multiple HiC data analysed or if invariable contacts like the SHH-ZRS shown are predominant. Shh has multiple enhancers controlling its expression in different tissues and developmental times - do the virtual 4C plots change in other data considered?

We have now prepared virtual 4C plots for Shh in additional conditions in human (promoter \pm 1.2Mb) and mouse (promoter \pm 1Mb). As can be seen (Supplementary Figure 10A, and also shown below). Putative interactions are identified for most mouse Hi-C experiments (top row, 850Kb), but not for most human SHH-ZRS (980Kb).

More generally, we have also analyzed the co-occurrence of putative enhancer regions found in GM12878 cells with other cell types (using a p-value threshold of $1e-10$) and computed the number of additional Hi-C experiments in which they are identified. These are shown in Supplementary Figure 10B (also below, blue bars). Putative DNA-DNA interactions tend to appear in multiple Hi-C conditions (compared to random, green, chi-square p-value $\leq 1e-300$).

4. The analysis is centred on a 2Mb region around promoters. Why not unbiasedly genome-wide to find potential enhancers even further away? The sentence on putative enhancers and their potential target genes should be made clearer (lines 218-220). How do the authors know the target promoter and what are the assumptions (proximity and single promoter?) ?

We focused our attention of regions interacting with gene promoters (e.g. enhancers) up to a certain distance, and 1Mb was chosen as an arbitrary number being the maximal distance of (currently well-studied) validated enhancers. Surely over-represented interactions could

be found at a larger distance, but with the risk of additional false positive calls. Currently, due to the relatively low resolution of Hi-C data, we resorted to assigning the putative enhancer region to every gene whose promoter falls within the enriched Hi-C cell. Additional analysis could be done to further pin-point the target genes and the accurate location of the putative enhancer, e.g. by intersecting these predicted enhancer regions with external data such as ChIP-seq of gene expression.

Minor points:

1. Inter-TAD merging was restricted to 5Mb but no justification given. Additionally it is a bit confusing that the paper considers inter-TAD interactions but BG is used in equations. Are these background or meaningful interactions?

As we show in Figure 4C, most (~88%) of the predicted DNA-DNA interactions occur with the same TAD, with additional 4% occurring within 1st-order TAD merges. The 5Mb threshold was therefore mainly used for efficiency reasons, with relatively few such interactions occurring inside inter-TAD regions likely to merged.

We have now rephrased the terms TAD and BG (in the context of the two-component model) to be more consistent with the rest of the paper (namely, by replacing them with “intra-TAD” and “inter-TAD”), and thank the reviewer for the clarification.

2. Using average DNA-DNA interaction as summary statistic for each domain can also lead to biases - not clear why the authors consider this better.

Our summary statistics of I_{intra} and I_{inter} are calculated for each distance d separately: $I_{\text{intra}}(d)$ and $I_{\text{inter}}(d)$. These are quite robust, as they are estimated from at least 10 Hi-C cells for each distance (otherwise set as NaN and ignored). In addition, this approach avoids the size-related biases favoring the merges of small TADs, compared to larger ones.

3. The two-component model is interesting but the justification given for using it should be better supported and explored. I think the reader would benefit from seeing these data plotted. Often there were Fig. S1 references where the plots did not correspond to expectations.

We have now elaborated on this matter. First, we have rephrased the legend for Supplementary Fig 1, to emphasize the it directly compares the empirical distribution of Hi-C DNA-DNA interactions (bold lines) vs. their log-Normal fitted model (dotted lines), for a variety of distances for both inter- and intra-TAD regions.

To further justify the log-Normal model, we have also compared the average likelihood (in \log_{10}) of the Hi-C data for both intra-TAD and inter-TAD interactions using other probabilistic families, including log-Poisson and Negative Binomial. These results are now shown as Supplemental Figure 1C-D, with a significantly better fit by a log-Normal model, thus supporting our model for the data representation.

Additional data are given in Supplementary Figure 1E supporting the robustness of our parameter estimation for the two-component model, as calculated on different chromosomes.

Reviewer #2 (Remarks to the Author):

The authors have provided a useful tool for determining significant interactions while identifying and taking into account topological domain structures. Some additional comparisons with existing datasets would be useful to further characterize the performance and robustness of the method. In addition, there are some improvements that could be made to the software provided that would increase its usability.

1. How much does the performance of the algorithm depend on the initial set of TADs, i.e. how robust is the method? The authors could address this point by showing how the final TAD calls are affected by using random TAD positions or a different initial TAD-calling algorithm.

We thank the reviewer for this comment, and have now compared the predictions of our PSYCHIC algorithm following three different initialization methods. In addition to the Directionality Index method (by Dixon et al), we have also implemented the Insulation Score method (Crane et al, Nature 2015), as well as random initialization of TADs. These changes had a limited effect on the predicted enhancer regions (using a p-value threshold of $1e-4$) on adult cortex Hi-C data from mouse chr10 (Dixon et al). These results are now shown as Supplementary Figure 6.

2. The manuscript doesn't state what binning resolution is used for any of the Hi-C datasets. This should be included.

We have now edited the text to include this information. We have used the original binning of each Hi-C experiments, except for Hi-C data from Rao et al, that were down-sampled to 25Kb resolution.

3. The methods should also include details of the Hi-C processing and format of input data.

We have now edited the methods section to include these details. Briefly, all Hi-C data we analyzed were already preprocessed and normalized by Hi-C normalization algorithms.

4. Figure 3: 4Mb windows around putative enhancers are far too large to assess enrichment of chromatin marks at the enhancers – this is more reflective of the enhancer bins being in active vs inactive compartments. This figure should be replaced with a higher resolution analysis to support the statement that the enhancer regions are enriched in active chromatin marks.

Supplemental Figure S4B now shows the data from Figure 3 in 1Mb windows. It should be noted that the analysis (for both Figure 3 and S4B) was done in the maximal resolution of the Hi-C data. Following the Reviewer's request, we have also included a statistical analysis of these data, by comparing the predicted enhancer location (center point) with distribution of average ChIP-seq values at distances $>500\text{Kb}$, to statistically quantify the sharpness of these plots. We thank the reviewer for the useful comment.

In addition, we have further emphasized the comparison with ChIP-seq signal at a random set of loci at a $<1\text{Mb}$ distance from gene promoters. As these comparisons show (Top of Fig

3, with heatmaps of random set now shown as Supplementary Figure 4A) the enrichment of the ChIP-seq signal (and others) are not due to active and inactive compartments (insignificant p-values for “random” control data).

5. It would be informative to know how well these enhancer-promoter interactions overlap with interactions determined using other types of data, such as ChIA-PET (e.g. PMID 26686651) or promoter capture Hi-C (e.g. PMID 25752748).

We thank the reviewer for raising this point, and have now analyzed the Capture Hi-C (CHi-C) data from Schoenfelder et al (Genome Res., 2015) as well as the ChIA-PET data from Tang et al (Cell, 2015). The Capture Hi-C data (in mESC) show high support for the predicted interactions by PSYCHIC, with coverage ranging from 69% of PSYCHIC interactions (FDR threshold of $1e-2$) to 74% (predictions at $FDR < 1e-4$), compared to 52%-66% for Fit-Hi-C predictions for mES Hi-C data (using thresholds of $1e-10$ and $1e-20$). These results are now shown as Supplementary Figure 9A.

In addition, we have analyzed GM12878 ChIA-PET data from the Ruan lab. The ChIA-PET interactions (PolII antibodies), show high support for our predictions, ranging from 37% (PSYCHIC GM12878 predictions with threshold of $p < 1e-2$) to 55% ($p < 1e-10$). HiCCUPS predictions for GM12878 yielded lower coverage by PolII ChIP-PET peaks. Intriguingly, a much higher portion of HiCCUPS calls (73%) were supported by CTCF ChIA-PET pairs, compared to only ~34% for PSYCHIC. This interesting result is in line with Figure 5, showing high ChIP-seq enrichments for CTCF (as well as Insulator marks) at putative enhancers called by HiCCUPS (and also Fit-Hi-C). These results are shown below and as Supplementary Figure 9B.

6. The authors report that detected enhancer-promoter interactions co-occur in a higher number of GAM slices than random interactions. What is the statistical significance of this increased co-occurrence?

Following the reviewer’s request, we have calculated the average GAM co-occurrence for 1,000 random shuffles of our enhancer-promoter set. The average number of slices for the

original set of predictions outperformed all 1000 shuffles, with a Normal distribution p-value of $5e-92$ (20.31 standard deviations above the average number of GAM slices in the shuffled set). We now report this statistical value in the text.

7. There is not much discussion of the features of the hierarchical TAD structure. Some examples of this structure at specific genomic loci would help readers understand the impact of the hierarchical merging. It would also be useful to include a comparison of this hierarchical approach with existing tools such as Armatus (PMID: 24868242).

We have now edited the text to consider the algorithms of Armatus (Filippova et al, 2014) and Fraser et al (2015). It should be noted that Armatus does not directly find a hierarchical structure, but instead identifies domains at different sizes (by altering the algorithm input parameters), thus generating multiple sets of TAD segmentations.

8. Software: I was unable to fully test the software, since I don't currently have access to a system with Matlab installed. However, I downloaded and ran the software on a system without Matlab installed, where it unexpectedly ran without producing any errors other than empty files with the results of the calculations. The usability of the software tool would be greatly increased by adding a few extra features:

- Checks for required software at the start of the run, with error messages/installation directions if necessary
- Usage/help information available using "--help" or "-h" or similar
- A description of the expected output files and their format
- A description of the required format for input files
- List required versions of all prerequisite software

We apologize for the inconvenience. Our method requires MATLAB (which is also required for the initialization of the two-component model, using the Directionality Index method by Dixon et al). The README file and the github software description have been improved, and now include a list of required software, as well as clear descriptions of the input file formats.

9. The authors refer to the bins detected as having enriched interactions with promoters as "enhancer regions" – a more neutral term that acknowledges the limited resolution of the Hi-C data would be more appropriate, such as "interacting bins" or "putative enhancer bins".

Fixed.

10. There are a few missing words / typos in the manuscript, e.g. line 12 "Proximity ligation methods such as Hi-C", line 255 "are possibly skewed by boundary elements"

We thank the reviewer for the comments. These typos were fixed.

Reviewer #3 (Remarks to the Author):

Ron et al present a new method to analyze a Hi-C (chromatin interaction) data set to reveal DNA-DNA interactions and then, by examining high confidence interactions with a promoter, they predict putative enhancers likely to regulate a gene. They provide circumstantial evidence in favor of the quality of the predicted enhancer collection, by showing enrichments of various enhancer-related properties. They also show that various enrichments are better than those seen when using state of the art methods for analyzing the same data to the same end (predicting enhancers). The paper is very well written.

We thank Reviewer 3 for their useful comments.

1. The finding shown in the last subsection (Figure 7) is exciting and intriguing. To me the statement “the hypothesis that a non-trivial fraction of the putative enhancer regions we have identified are “silent” and inaccessible. A closer examination identified several known enhancers even within those” was the most interesting aspect of the entire paper, and I was wishing that the authors had made the point more forcefully (e.g., by at least providing some basic numbers and listing more examples like ZRS-Shh in the supplement or even in main text), to make readers think of the possibilities here, even if there is the usual risk of false positives.

We have now prepared virtual 4C plots for Shh in additional conditions in human and mouse. As can be seen (Supplementary Figure 10A, and also below), putative interactions are identified for most mouse Hi-C experiments (top row, ZRS located 850Kb from Shh), but not for most human SHH-ZRS (980Kb apart from SHH).

A full scale analysis of the conservation of promoter-enhancer interactions (in inactive conditions) is rather premature, due to the limited availability of validated promoter-enhancers interactions.

2. To what extent is the difference in predicted enhancers between the new method and previous methods due to difference in segmentation versus difference in how intra and inter-TAD interaction intensities are normalized to predict significant interactions? I am guess that the segmentation may not be very different and that the main differences seen in Figs 5 and 6 arise from the normalization procedure.

Indeed, we believe that the strength of PSYCHIC lies mostly in the TAD-specific background model, which allows for a better identification of interactions regardless of the general tendency for interactions in each TAD.

To further support this hypothesis, we have repeated the comparison with different TAD calling methods for all chromosomes, as shown in Supplementary Figure 2 (please note that we also fixed a minor bug in the Y-axis scaling). Here, we compare no TAD segmentation at all (that is, a single background model for each chromosome) as shown as the 1st bar of Supplementary Figure 2. Indeed, this results with a much poorer fit (bars show the average RMSE, with error-bars corresponding to 25th and 75th percentiles). Segmentation into random TADs (random shuffles of called TAD), as well as TAD called using DI (Dixon et al), allow for less accurate fit compared compared to the PSYCHIC TADs, with or without hierarchical and bilinear modes (Supplementary Figure 2).

In addition, we have compared the effect of initialization by different TAD calling methods, now compare the Directionality Index method (Dixon et al) to the Insulation Square (Crane et al, Nature 2015), as well as initialization by random TADs. These changes had a limited effect on the predicted enhancer regions (using a p-value threshold of $1e-4$) on adult cortex Hi-C data from mouse chr10 (Dixon et al). These results are now shown as Supplementary Figure 6.

3. The idea of subtracting the expectation can be a double edged sword in this application. In comparing the residuals of different windows (when constructing the Normal distribution), the assumption is that enhancers may be present at all distances with equal prior probability. If in reality there is a much greater crowding of enhancers closer to (say within 100 Kb of) the gene, then the Hi-C data will persistently show strong intensities and the background model might discount that stronger signal. I am not sure what it does to the signal/noise ratio, but it clearly hurts the sensitivity of the method, in a way biased against closer enhancers.

We completely agree with the Reviewer but are hesitant at this stage to add a bias (which we cannot strongly support) towards promoter-enhancer interactions at given distances. Many known enhancers were identified thanks to their proximity after “searching under the streetlight”. As figure 4A shows, the decay in putative enhancer-promoter interactions is rather slow, suggesting the presence of distal enhancers within (or possible even outside of) the same TAD as their target genes.

4. The score $S(t)$ is somewhat intriguing since it is a heuristic as far as I can tell. If the “outside of TAD t ” pairs always involve going a fixed distance (h) outside of the TAD, then the number of such “outside” pairs scales linearly with the TAD length and the number of inside pairs scales quadratically with TAD length. What effect does this have on $S(t)$ and are these values comparable across different domains ?

We thank the Reviewer for their interest, would like to expand on this point which we have carefully designed. The score of each TAD serves as the basic unit in the Dynamic Programming algorithm, allowing for a computationally-efficient segmentation of one region into two (Equation 5). Formally, score $S(t)$ is the sum of log-posterior ratios for the entire rectangle above the TAD t , composed of the triangle intra-TAD cells at the base, as well as the inter-TAD interactions above it (see Figure 1C), with the important distinction

that intra-TAD cells are counted with a positive sign (posterior of intra-TAD being in the numerator of the log), while inter-TAD are negative (denominator).

In the Dynamic Programming algorithm, when comparing the Score of one vs two segments, the overall region considered (by both alternatives) is equal, and only the “merged” cells (striped region in Figure 1C) have different signs - according to the one and two-TAD models, considered intra-TAD by the former, and inter-TAD by the latter. Usefully, this only changes the sign of their log-posterior ratio score. So when considering this possible segmentation, the score of other cells cancel out, and those Merge cells (= tilted rectangle in between) get to “vote” if they are more likely to have been generated from a single TAD (=their added scores are greater than zero) or from two distinct TADs (=negative sum). This is also why we capped the maximal allowed distance for DNA-DNA interactions at 5Mb so that the tilted rectangle will not be capped at the top corner.

5. The score ‘Score(C)’ (equation 8), in adding $S(t)$ over multiple (non-overlapping) TADs appears to include the negative part of equation(7) RHS potentially multiple times. Am I missing something?

We hope the previous answer and the elaborated description in the text have now cleared this issue. Generally, all $S(t)$ s are rectangles with the same height h , and overlap is not possible.

6. The motivation and practical utility of the hierarchical structure formation is not entirely clear. While it is clear that visualization of the HiC maps suggests the presence of such structures, how exactly do the authors propose the user of this tool will utilize the hierarchical structure?

The main scope of the PSYCHIC algorithm is to identify promoter-enhancer interactions. A better segmentation into TAD, together with the inter-TAD hierarchical structure serve as a means for a better fit of the Hi-C data, thus allowing for a better identification of promoter-enhancer interactions. As we have showed in Supplementary Figure 2 TAD-specific background models allow for a more accurate modeling of the Hi-C data. This is further improved with hierarchical and moreover by the bilinear fit models.

7. Suppl Figure 2 needs error bars, possibly based on bootstrap sampling, for each bar. (No need for p-values, standard errors for each estimate will be just fine.)

We thank the reviewer for this idea and have repeated this analysis for all chromosome (with error-bars corresponding to the 25th and 75th percentile values in each group). We have also fixed a minor bug in the Y-axis scaling.

8. I am not entirely clear why the fourth from left bar in Supplementary Figure 2 is called 'Hierarchical TADs'. I saw that the description of the model talks about hierarchical agglomeration of neighboring domains, followed by local parameter estimation. What is not clear is why the agglomeration had to be done to assign TAD (and inter-TAD) specific parameters for modeling local intensity. Couldn't the same exact result be obtained just from the knowledge of all TADs (the primary segmentation) followed by local parameter assignments? Or is there some sort of threshold applied to the hierarchical organization that determines which inter-TADs will be considered and which not. In short, I am not clear on why the hierarchical structure was needed in this step.

We apologize for the confusion and have edited the new Supplementary Figure 2. Briefly, it now compares the RMSE of all Hi-C cells (up to 5Mb), averaged over all chromosomes of (from left to right): (1) no TAD segmentation (as in Fit-Hi-C); (2) TAD-specific fit for random (shuffled) domains; (3) TAD-specific fit for Dixon et al domains; (4) TAD-specific fit for PSYCHIC-called domains; (5) fit for PSYCHIC domains and inter-TAD merges (as in Figure 1, where TADs A and B are merged, while TAD C is not); (6) fit for PSYCHIC TADs (4th column), now with TAD-specific bi-linear fit model; (7) fit for hierarchical TADs (5th column) now with bi-linear fit model; and (8) fit for synthetic data (with similar Hi-C magnitude to real-life data) using the same TADs (and merges) that had generated it, to reflect the internal (sampling) noise in the data.

9. How different would the putative enhancer set be if using the simple power law rule for computing residuals in each promoter's 4 Mb context, followed by the same z-score / p-value calculation, but without utilizing the hierarchical structure, and even the TAD segmentation? I would expect that the difference is likely not major for within-TAD interactions, and any significant differences are probably in inter-TAD regions, which hosts a small minority (~10%) of putative enhancers anyways.

We have now updated Supplementary Figure 2 (shown above) to also show the fit obtain with no TADs at all (namely, a single power-law model for each chromosome). As can be seen the effect is quite dramatic, and is due to both inter-TAD interaction but also due to

intra-TAD interactions which benefit by different background models for different TADs (e.g. in “dark” and “pale” TADs).

10. Figure 5 provides evidence that the predicted putative enhancers might be more reliable than those predicted by other methods on the same Hi-C data. However, the height of each bar (in each panel) is the ‘average signal’ of a certain type in the corresponding collection of predicted enhancers, and leaves open the possibility that there is a precision-recall tradeoff in play here, and that the average signal comparisons might look different at alternative values of cutoffs used for the other methods. I noticed that the authors have tried to keep this in mind in the evaluations of Figure 6A, where they show higher precision even with greater numbers of predictions made by their method.

Indeed, when possible we have tried to provide several thresholds, such that the methods could be compared fairly. HiCCUPS did not rank their predictions.

11. While the authors have chosen to validate their predictions using various other sources of enhancer-related information, and while these are helpful in making their point, these do not quite reach the level of ‘validation’. I would have been much more convinced if the authors had pointed out, for instance, that one or both of the ‘true’ *Foxg1* enhancers would not have been picked by a serious scientist who did not have access to the method proposed here. A more systematic evaluation might be based on enhancer databases such as VISTA enhancer browser.

The VISTA dataset is somewhat limited in the scope of positive enhancers, as all validations are in embryonic mice around E10.5. For example, 342 out of 610 positive mouse enhancers (56%) overlap a putative PSYCHIC enhancer, but they only make ~2% of the 17,788 predicted regions by PSYCHIC at a threshold of $1e-2$).

Instead, we have now compared our predictions to Capture Hi-C (mESC, Schoenfelder et al. Genome Res., 2015) and ChIA-PET (GM12878, Tang et al. Cell, 2015). As we show in Supplementary Figure 9 (also below), PSYCHIC predictions are strongly supported by these external enhancer-promoter datasets.

Intriguingly, HiCCUPS predictions are noticeably enriched by CTCF ChIA-PET interactions, a result consistent with Figure 5, where we showed high ChIP-seq enrichments for CTCF (as well as the “Insulator” mark) at putative enhancers called by HiCCUPS, possibly by their bias towards boundary-related interactions.

12. The finding shown in the last subsection (Figure 7) is exciting and intriguing. To me the statement “the hypothesis that a non-trivial fraction of the putative enhancer regions we have identified are “silent” and inaccessible. A closer examination identified several known enhancers even within those” was the most interesting aspect of the entire paper, and I was left wishing that the authors had made the point more forcefully (e.g., by at least providing some basic numbers and listing more examples like ZRS-Shh in the supplement or even in main text), to make readers think of the possibilities here, even if there is the usual risk of false positives.

Already addressed above (point #1)

13. Line 124: “all inter-TAD outside of TAD t, defined by pairs ” This needs some rewording/correction. You mean “all inter-TAD pairs outside of TAD t, defined by pairs $\langle k, l \rangle$ ” ?

We thank Reviewer 3. Indeed our intention was “all inter-TAD pairs outside of TAD t. We have now reworded this entire section.

14. Line 141: “Specifically, we wish to iteratively agglomerative neighboring” Change ‘agglomerative’ to ‘agglomerate’ ?

Fixed

15. Line 169: “significance over-represented interactions” Missing “of” ?

Fixed

16. Line 155: “Once we have segmented the Hi-C map into hierarchical domains, we wish to model the expected intensity of the Hi-C map”. At this point, the authors should provide a brief motivation of where they want to go with this. Very related point: I was for some time confused by the intent of Supplementary Figure 2, because it seemed to show that more parameterized models give better fits to data, which is completely expected. The authors put my doubts to rest with their last sentence in the subsection, that the goal is to build a more accurate background model to assess deviations from. This is of course a perfectly reasonable strategy and the authors should note it up front.

We have now rephrased this section and thank the Reviewer.

REVIEWERS' COMMENTS:

Reviewer #1 (Remarks to the Author):

The authors have successfully address most of the points raised, except that they have still not done a good job at describing their hierarchies. This issue was also raised by another reviewer.

They state: "maximal merge size of 5Mb, to create a set (forest) of tree-like TAD merges, visually corresponding to triangles (TADs) and rectangles (inter-TAD merge regions)."

But Fig1 only shows two TADs A and B being merged. Are all their merges level 1? Or in the 5Mb cutoff they also found instances of multiple (>2), small TADs being hierarchically merged? It seems that this is the case, since they call this set tree-like merges but we are not told/given this information.

The authors should provide this last clarification, add an histogram with this info and one sentence to explain this better.

Reviewer #2 (Remarks to the Author):

Most of our points have been successfully addressed by the authors. In particular, it is good to see that the method seems to be robust to different initial sets of TADs, and that the predicted interactions are well supported by physical interaction data. The additional methods clarification is also sufficient. However, there are a couple of remaining issues.

1. The authors have included a new higher resolution analysis of the characteristics of the identified putative enhancer regions, but are limited by the resolution of the Hi-C data. Given the increasing resolution of published Hi-C datasets (e.g. the Rao et al. data is available at 5kb but analysed at 25kb), I think it's highly likely that readers would be interested in the application of this method to higher resolution datasets. It would be helpful if the authors could include some discussion of the limitations of using this method with higher resolution data, e.g. scalability.

2. Not all of the suggestions relating to the software have been implemented, which is a shame as I think these would make the software more user-friendly and therefore more likely to be widely used. Help / usage information has been added, which is good. The list of required software is adequate but I'm still surprised that the software does not test for these dependencies at the start of the analysis. While providing an example file for reference is great, "should be in either CSV file or Bing Ren & Dixon format" is not a sufficient description for the input file - there are many ways to represent Hi-C data in a CSV file... There is also still no list of expected output files and their contents. This would be very helpful for the user to verify that the software has completed successfully and to know which files to use for downstream analysis.

Reviewer #3 (Remarks to the Author):

The authors have done a sincere job at addressing the concerns and questions raised in my review. In most cases they have presented additional insights to clarify points of confusion. In the cases where they were unable to directly address a request, I am not going to insist further since I understand their point of view in those cases, and feel it is up to their discretion.

Point-by-point response for Ron et al, "Promoter-Enhancer Interactions Identified from Hi-C Data using Probabilistic Models and Hierarchical Topological Domains", NCOMMS-17-11126A.

Reviewer #1 (Remarks to the Author):

The authors have successfully address most of the points raised, except that they have still not done a good job at describing their hierarchies. This issue was also raised by another reviewer. They state: "maximal merge size of 5Mb, to create a set (forest) of tree-like TAD merges, visually corresponding to triangles (TADs) and rectangles (inter-TAD merge regions)."

But Fig1 only shows two TADs A and B being merged. Are all their merges level 1? Or in the 5Mb cutoff they also found instances of multiple (>2), small TADs being hierarchically merged? It seems that this is the case, since they call this set tree-like merges but we are not told/given this information.

The authors should provide this last clarification, add an histogram with this info and one sentence to explain this better.

Figure 1d was edited to highlight the possibility of further hierarchies (dotted lines; clarification sentence). We have also added Supplementary Figure 1g per Reviewer 1's request, specifying the number of topological domains from each level (basic TADs through 4th order merges).

Reviewer #2 (Remarks to the Author):

Most of our points have been successfully addressed by the authors. In particular, it is good to see that the method seems to be robust to different initial sets of TADs, and that the predicted interactions are well supported by physical interaction data. The additional methods clarification is also sufficient. However, there are a couple of remaining issues.

1. The authors have included a new higher resolution analysis of the characteristics of the identified putative enhancer regions, but are limited by the resolution of the Hi-C data. Given the increasing resolution of published Hi-C datasets (e.g. the Rao et al. data is available at 5kb but analysed at 25kb), I think it's highly likely that readers would be interested in the application of this method to higher resolution datasets. It would be helpful if the authors could include some discussion of the limitations of using this method with higher resolution data, e.g. scalability.

A discussion on the computational limitation and scalability of the method was added.

2. Not all of the suggestions relating to the software have been implemented, which is a shame as I think these would make the software more user-friendly and therefore more likely to be widely used. Help / usage information has been added, which is good. The list of required software is adequate but I'm still surprised that the software does not test for these dependencies at the start of the analysis. While providing an example file for reference is great, "should be in either CSV file or Bing Ren & Dixon format" is not a sufficient description for the input file – there are many ways to represent Hi-C data in a CSV file... There is also still no list of expected output files and their contents. This would be very helpful for the user to

verify that the software has completed successfully and to know which files to use for downstream analysis.

We thank the reviewer. We now test for dependencies and installed software. In addition we have supplied in-house alternatives for the external code we have initially used (e.g. Dixon et al's Directionality Index). Following Reviewer 2's comment, we have clarified the format's description and a description of the output files.

Reviewer #3 (Remarks to the Author):

The authors have done a sincere job at addressing the concerns and questions raised in my review. In most cases they have presented additional insights to clarify points of confusion. In the cases where they were unable to directly address a request, I am not going to insist further since I understand their point of view in those cases, and feel it is up to their discretion.

We thank Reviewer 3 for their comments and support.